# TENCENT | GPR

# PRISM: Parallel Residual Iterative Sequence Model

Jie Jiang [*1]  Ke Cheng [*12]  Xin Xu [*13]  Mengyang Pang [*1]  Tianhao Lu [*1]  Jiaheng Li [3]  Yue Liu [1]  Yuan Wang [1]  Jun Zhang [✉1]  Huan Yu [1]  Zhouchen Lin [✉3]

## Abstract

Generative sequence modeling faces a fundamental tension between the expressivity of Transformers and the efficiency of linear sequence models. Existing efficient architectures are theoretically bounded by shallow, single-step linear updates, while powerful iterative methods like Test-Time Training (TTT) break hardware parallelism due to two dimensions of serial dependency: token-level state reliance and step-level iteration loops. We propose PRISM (Parallel Residual Iterative Sequence Model) to resolve this tension. PRISM explicitly reconstructs the expressive gate × residual × direction iteration pattern of TTT in a parallelizable form. We employ a Write-Forget Decoupling strategy that isolates non-linearity within the injection operator. To bypass the serial dependency of explicit solvers, PRISM utilizes a two-stage proxy architecture: a short-convolution anchors the initial residual using local history energy, while a learned predictor estimates the refinement updates directly from the input. This design distills structural patterns associated with iterative correction into a parallelizable feedforward operator. Theoretically, we prove that this formulation achieves Rank-$L$ accumulation, structurally expanding the update manifold beyond the single-step Rank-1 bottleneck. Empirically, it achieves comparable performance to explicit optimization methods while achieving **174x higher throughput**. Codes are available in https://github.com/gpr-prism/prism/.

---

[*]Equal contribution  [1]Tencent AMS., Beijing, China  [2]State Key Laboratory of software development environment, School of Computer Science and Engineering, Beihang University, Beijing, China  [3]State Key Laboratory of General Artificial Intelligence, School of Intelligence Science and Technology, Peking University, Beijing, China. Correspondence to: Jun Zhang <neoxzhang@tencent.com>, Zhouchen Lin <zlin@pku.edu.cn>.

*Proceedings of the 43rd International Conference on Machine Learning*, Seoul, South Korea. PMLR 306, 2026. Copyright 2026 by the author(s).

## 1. Introduction

Generative sequential modeling (e.g., HSTU (Zhai et al., 2024)) has revolutionized recommender systems by unlocking scaling laws for long-term user interests (Deng et al., 2025). However, the Transformer's quadratic complexity ($O(N^2)$) makes modeling lifelong sequences prohibitively expensive (Vaswani et al., 2017). Current solutions often resort to lossy heuristics like sliding windows or static compression to reduce costs, inevitably severing critical dependencies. Consequently, developing an efficient architecture capable of handling ultra-long sequences without sacrificing fidelity remains a key challenge.

Recent work has established that sequence modeling can be unified under an *Online Convex Programming* (OCP) framework (Yang et al., 2024b; Sun et al., 2024): at each step $t$, the model solves a proximal optimization problem to update its memory state $\mathbf{S}_t$. Under this lens, efficient architectures such as Linear Attention (Yang et al., 2024b;a), SSMs (Gu et al., 2020; Gu & Dao, 2024), and the Delta Rule family (Schlag et al., 2021; Yang et al., 2024b) all correspond to *linearized, one-shot* solutions to this objective. This linearization yields two desirable properties—closed-form updates and compatibility with parallel scan—but imposes a fundamental *Rank-1 Bottleneck*: each token's contribution to memory is structurally confined to a single outer product $\mathbf{v}_t \mathbf{k}_t^\top$, limiting the model's capacity to capture high-order feature interactions or iteratively minimize reconstruction error (Lei et al., 2025).

Explicit optimization methods such as Test-Time Training (TTT) (Zhang et al., 2025; Behrouz et al., 2025b) resolve the rank bottleneck by performing multi-step non-linear gradient descent on the *same* OCP objective at runtime, achieving Rank-$L$ updates. However, this comes at the cost of the *Serial Dependency Bottleneck*: since each gradient step depends on the updated state from the previous step, the optimization loop is inherently sequential, breaking hardware parallelism and negating the throughput advantages of efficient models (Yang et al., 2023; Behrouz et al., 2024).

We observe that both families optimize the *same* underlying objective—a proximal term plus a non-linear retrieval loss $\|\mathbf{S}_t - \alpha_t \mathbf{S}_{t-1}\|_F^2 + \beta_t \|\sigma(\mathbf{S}_t \mathbf{k}_t) - \mathbf{v}_t\|^2$—but differ in *how* they solve it. One-shot methods linearize and accept Rank-

1; TTT keeps the non-linearity but sacrifices parallelism. This motivates a natural question: *can we achieve Rank-L updates while preserving parallel scan compatibility?*

We answer affirmatively with **PRISM** (**P**arallel **R**esidual **I**terative **S**equence **M**odel). Our key insight is that while the exact gradient trajectory of the non-linear objective is state-dependent, its *structure*—specifically, the gate-residual-direction decomposition that emerges from multi-step optimization—can be amortized into a learned static policy conditioned solely on the input. This allows us to emulate the functional benefits of deeper optimization (Rank-$L$ accumulation and non-linear shaping) as a parallelizable structural inductive bias.

To operationalize this, PRISM employs a **Write-Forget Decoupling** strategy: we maintain hardware-efficient linear decay for forgetting dynamics (identical to Gated DeltaNet) while isolating the expressivity gains within the injection operator. To bypass the serial bottleneck, we introduce **Input-Anchored Loop Unrolling**: a two-stage proxy mechanism that (1) anchors the initial residual via short-convolution capturing local history energy, and (2) applies $L$ learned direction-gate pairs $\{(\mathbf{k}^{(l)}, \mathbf{p}^{(l)})\}_{l=1}^{L}$ to produce a Rank-$L$ update in a single fused pass. The entire refinement loop collapses into a closed-form parallel operator.

Our main contributions are:

- **Unified OCP Perspective with Rank-Parallel Taxonomy.** We position existing methods within a common optimization framework (Table 1) and identify the Rank-1/Parallel trade-off as the central bottleneck. This clarifies that PRISM solves the *same* objective as TTT and Gated DeltaNet, but is the first to achieve both Rank-$L$ and parallel scan compatibility.

- **PRISM Architecture.** We propose a hardware-aware implementation via Write-Forget Decoupling and Input-Anchored Loop Unrolling. The two-stage proxy mechanism circumvents the inherent serial computation bottleneck of iterative solvers while preserving their expressive power. Empirically, PRISM attains performance on par with TTT while achieving throughput improvements of up to **174×**.

- **Theoretical Expressivity Guarantee.** We formally prove that the input-anchored formulation yields *Rank Accumulation*: PRISM's injection operator produces Rank-$L$ state modifications per token, asymptotically approaching the solution quality of the ideal non-linear delta rule. We further show that when $L=0$ (no solver steps), PRISM structurally reduces to Gated DeltaNet, establishing backward compatibility.

## 2. Related Work

We survey efficient sequence modeling through the lens of **Online Convex Programming** (OCP) (Yang et al., 2024b; Sun et al., 2024), which provides a unified optimization-theoretic framework for understanding linear recurrences. Under this framework, all methods can be characterized by (1) how they formulate the per-step objective, and (2) how they solve it—yielding a natural taxonomy along two axes: **Rank** (expressivity of the per-token update) and **Parallel Scan compatibility** (hardware efficiency). A comparison is provided in Table 1.

### 2.1. One-Shot Linear Solutions (Rank-1, Parallel)

The dominant approach for efficient sequence modeling constrains the state update to a single-step, closed-form solution of a linearized proximal objective. Under the OCP framework, this family arises from dropping the non-linear activation $\sigma$ and solving in one step.

**Additive and Gated Updates.** Linear Attention (Katharopoulos et al., 2020) and RWKV (Peng et al., 2025) employ a pure Hebbian accumulation $\mathbf{S}_t = \mathbf{S}_{t-1} + \mathbf{v}_t \mathbf{k}_t^\top$. Mamba (Gu & Dao, 2024) and Mamba-2 (Dao & Gu, 2024) introduce input-dependent gated decay $\alpha_t$, improving context filtering but retaining a Rank-1 injection. GLA (Yang et al., 2023) and GSA (Zhang et al., 2024) further enhance the gating mechanism.

**Delta Rule Family.** DeltaNet (Yang et al., 2024b) formulates the update as a one-step gradient on a linearized retrieval loss, yielding the error-correcting form $\mathbf{S}_t = \mathbf{S}_{t-1} \cdot (\mathbf{I} - \beta_t \mathbf{k}_t \mathbf{k}_t^\top) + \beta_t \mathbf{v}_t \mathbf{k}_t^\top$. Gated DeltaNet (Yang et al., 2024a) combines this with scalar decay $\alpha_t$. PGDN (Tumma et al., 2026) further improves convergence by changing the norm of the proximal term via a preconditioner $\mathbf{P}$, rotating the key direction to $\tilde{\mathbf{k}}_t = \mathbf{P}\mathbf{k}_t$. All members of this family are structurally Rank-1 and fully compatible with parallel scan, as the linearization preserves the matrix semi-ring structure required for associative prefix operations.

### 2.2. Multi-Step Non-Linear Solutions (Rank-$L$, Serial)

**Test-Time Training (TTT).** TTT (Sun et al., 2024; Zhang et al., 2025) resolves the Rank-1 bottleneck by performing $L$ steps of gradient descent on the *full* non-linear objective $\|\sigma(\mathbf{S}_t \mathbf{k}_t) - \mathbf{v}_t\|^2$ at runtime. Each step produces an independent gradient direction, yielding up to Rank-$L$ state modifications. TTT-NN further uses an MLP state to implicitly provide multi-direction via its hidden layer $W_1$. However, because each gradient step $\nabla_{\mathbf{S}} \mathcal{L}$ depends on the current state $\mathbf{S}^{(l)}$, the optimization loop is inherently sequential—each step must wait for the previous step's state update. This **Serial Dependency Bottleneck** precludes parallel scan and forces $O(NL)$ sequential computation during

*Table 1.* **Linear recurrences as online learning.** All methods optimize a proximal objective with varying fidelity. The *Rank* column indicates the rank of the per-token state modification. PRISM is the first to achieve both Rank-$L$ and parallel scan compatibility by amortizing the multi-step solution via input-anchored proxies. Table structure follows Tumma et al. (2026).

| Method | Online Learning Objective | Rank | Par. Scan |
|---|---|---|---|
| LA (Katharopoulos et al., 2020) | $\|\mathbf{S}_t - \mathbf{S}_{t-1}\|_F^2 - 2\langle\mathbf{S}_t\mathbf{k}_t, \mathbf{v}_t\rangle$ | 1 | ✓ |
| Mamba-2 (Dao & Gu, 2024) | $\|\mathbf{S}_t - \alpha_t\mathbf{S}_{t-1}\|_F^2 - 2\langle\mathbf{S}_t\mathbf{k}_t, \mathbf{v}_t\rangle$ | 1 | ✓ |
| DeltaNet (Yang et al., 2024b) | $\|\mathbf{S}_t - \mathbf{S}_{t-1}\|_F^2 - 2\langle\mathbf{S}_t\mathbf{k}_t, \beta_t(\mathbf{v}_t - \mathbf{S}_{t-1}\mathbf{k}_t)\rangle$ | 1 | ✓ |
| Gated DeltaNet (Yang et al., 2024a) | $\|\mathbf{S}_t - \alpha_t\mathbf{S}_{t-1}\|_F^2 - 2\langle\mathbf{S}_t\mathbf{k}_t, \beta_t(\mathbf{v}_t - \alpha_t\mathbf{S}_{t-1}\mathbf{k}_t)\rangle$ | 1 | ✓ |
| PGDN (Tumma et al., 2026) | $\|\mathbf{S}_t - \alpha_t\mathbf{S}_{t-1}\|_{\mathbf{P}^{-1}}^2 - 2\langle\mathbf{S}_t\mathbf{k}_t, \beta_t(\mathbf{v}_t - \alpha_t\mathbf{S}_{t-1}\mathbf{k}_t)\rangle$ | 1 | ✓ |
| TTT (Sun et al., 2024) | $\|\mathbf{S}_t - \alpha_t\mathbf{S}_{t-1}\|_F^2 + \beta_t\|\sigma(\mathbf{S}_t\mathbf{k}_t) - \mathbf{v}_t\|^2$ | up to $L$ | ✗ |
| **PRISM (Ours)** | $\|\mathbf{S}_t - \alpha_t\mathbf{S}_{t-1}\|_F^2 + \beta_t\|\sigma(\mathbf{S}_t\mathbf{k}_t) - \mathbf{v}_t\|^2$ | up to $L$ | ✓ |

training, making TTT up to $174\times$ slower than linear recurrences at equivalent scale.

**Titans** (Behrouz et al., 2024) extends TTT with a momentum-based memory and surprise-gated updates, but fundamentally shares the same serial constraint as the optimization loop remains state-dependent.

### 2.3. Positioning PRISM: Amortized Multi-Step Solution (Rank-$L$, Parallel)

PRISM resolves the tension between expressivity and efficiency by targeting the *same* non-linear objective as TTT—$\|\mathbf{S}_t - \alpha_t\mathbf{S}_{t-1}\|_F^2 + \beta_t\|\sigma(\mathbf{S}_t\mathbf{k}_t) - \mathbf{v}_t\|^2$—while *amortizing* the multi-step solution to eliminate serial dependencies.

The key observation is that TTT's seriality arises from two sources: (1) the residual $\mathbf{r}^{(l)} = \mathbf{v}_t - \sigma(\mathbf{S}^{(l)}\mathbf{k}_t)$ depends on the evolving state, and (2) the gradient direction changes with $W_1^{(l)}$ after each weight update. PRISM breaks both dependencies simultaneously:

- **Residual**: approximated by a local *input-anchored* proxy computed from a short convolution over the input sequence, removing the dependency on $\mathbf{S}_{t-1}$.

- **Direction**: replaced by $L$ *learned projections* precomputed from the same local proxy, eliminating inter-step dependency.

This yields a Rank-$L$ injection $\mathbf{B}_t = \sum_{l=1}^{L} \beta^{(l)}\boldsymbol{\delta}^{(l)}\mathbf{k}^{(l)\top}$ that is fully parallelizable while structurally emulating the multi-step refinement trajectory. When $L=1$, PRISM structurally reduces to Gated DeltaNet, establishing it as a strict generalization of the Rank-1 family.

## 3. Theoretical Motivation

In this section, we establish a unified optimization-theoretic view of sequence modeling, identify the fundamental Rank-Parallel trade-off that governs existing architectures, and

motivate PRISM as a resolution.

### 3.1. Sequence Modeling as Online Convex Programming

Notation. At time step $t$, let $\mathbf{x}_t$ denote the input representation, $\mathbf{S}_t \in \mathbb{R}^{d_v \times d_k}$ the recurrent state, $\mathbf{k}_t, \mathbf{q}_t \in \mathbb{R}^{d_k}$ the key and read query, and $\mathbf{v}_t \in \mathbb{R}^{d_v}$ the value. We use the right-multiplication convention $\mathbf{S}_t = \mathbf{S}_{t-1}\mathbf{A}_t + \mathbf{B}_t$, where $\mathbf{A}_t \in \mathbb{R}^{d_k \times d_k}$ and $\mathbf{B}_t \in \mathbb{R}^{d_v \times d_k}$. Thus, $\widehat{\mathbf{v}}_t = \mathbf{S}_t\mathbf{q}_t$. Bold uppercase letters denote matrices or sequence tensors, bold lowercase letters denote vectors, and unbolded letters denote scalars, indices, or dimensions. We set $d_k = d_v = d$ in the configurations used in this paper.

**The Unified Objective.** Following Yang et al. (2024b); Sun et al. (2024), we view state updates in linear recurrences as solutions to a per-step online optimization problem. At each timestep $t$, the model seeks a state $\mathbf{S}_t \in \mathbb{R}^{d \times d}$ that balances two competing goals: (1) *stability*—the new state should not deviate too far from the previous one, and (2) *fidelity*—the state should accurately retrieve the current value $\mathbf{v}_t$ when queried with key $\mathbf{k}_t$. This is formalized as:

$$\mathcal{L}_t(\mathbf{S}) = \frac{1}{2}\|\mathbf{S} - \alpha_t\mathbf{S}_{t-1}\|_F^2 + \frac{\lambda_t}{2}\|\phi(\mathbf{S}\mathbf{k}_t) - \mathbf{v}_t\|_2^2. \quad (1)$$

where $\alpha_t \in (0, 1]$ is a scalar decay gate, $\lambda_t > 0$ weights retrieval fidelity, and $\phi$ is an element-wise activation. We use sigmoid$(\cdot)$ exclusively for logistic gates. The identity matrix $\mathbf{I}$ has shape $d_k \times d_k$. This objective encompasses the entire spectrum of existing methods depending on how it is solved (see Table 1).

**One-Shot Linear Solution $\Rightarrow$ Rank-1.** For the linearized case, initialize $\mathbf{S}_t^{(0)} = \mathbf{S}_{t-1} \cdot \alpha_t$. One delta update with $\beta_t^{(0)} = \eta_t\lambda_t$ gives

$$\mathbf{S}_t^{\text{base}} = \mathbf{S}_{t-1} \cdot \alpha_t \left( \mathbf{I} - \beta_t^{(0)} \mathbf{k}_t^{(0)} \mathbf{k}_t^{(0)\top} \right) + \beta_t^{(0)} \mathbf{v}_t \mathbf{k}_t^{(0)\top}. \tag{2}$$

which is exactly the Gated DeltaNet update (Yang et al., 2024a). The additive base write $\beta_t^{(0)} \mathbf{v}_t \mathbf{k}_t^{(0)\top}$ is rank 1 in each head. This structure preserves the matrix semi-ring form required for parallel prefix scans, enabling $O(\log N)$ parallel computation.

**The Rank-1 Bottleneck.** While efficient, the Rank-1 constraint means that a single token can only "write" information along one key direction per step. To capture multi-faceted associations (e.g., a token that is simultaneously relevant to multiple queries), the model must rely entirely on depth—stacking multiple layers—rather than within-step refinement. This fundamentally limits per-layer expressivity.

### 3.2. The Non-Linear Solver

The linearized update in Eq. (2) obtains a scan-compatible base write by replacing the nonlinear response with an identity mapping. To recover response-dependent adaptation, we retain the nonlinear $\phi$ in Eq. 1 and consider iterative optimization. where $\phi'(\cdot)$ denotes the element-wise derivative. A single gradient step yields the *Non-Linear Delta Rule*:

$$\mathbf{S}_t^{(1)} = \mathbf{S}_t^{(0)} + \eta_t \lambda_t \left[ \phi'(\mathbf{S}_t^{(0)} \mathbf{k}_t) \odot \left( \mathbf{v}_t - \phi(\mathbf{S}_t^{(0)} \mathbf{k}_t) \right) \right] \mathbf{k}_t^{\top}, \tag{3}$$

where $\mathbf{S}_t^{(0)} = \alpha_t \mathbf{S}_{t-1}$. For notational consistency, let $\beta_t^{(0)} = \eta_t \lambda_t \phi'(\mathbf{S}_t^{(0)} \mathbf{k}_t)$, $\boldsymbol{\delta}_t^{(0)} = \mathbf{v}_t - \phi(\mathbf{S}_t^{(0)} \mathbf{k}_t)$, and $\mathbf{k}_t^{(0)} = \mathbf{k}_t$. Equation (3) can then be written as $\Delta \mathbf{S}_t = \beta_t^{(0)} \boldsymbol{\delta}_t^{(0)} \mathbf{k}_t^{(0)\top}$. Here, $\beta_t^{(0)}$ is a scalar step coefficient, $\boldsymbol{\delta}_t^{(0)}$ is a gain-modulated residual correction, and $\mathbf{k}_t^{(0)}$ is the write direction. This update is *adaptive*: in saturated memory regions where $\phi'$ is small, the correction is automatically suppressed; in sensitive regions where the memory response lies in the linear range of $\phi$, the correction is larger. However, a single step still produces only a Rank-1 additive update along $\mathbf{k}_t^{(0)}$.

**Multi-Step $\Rightarrow$ Rank-$L$.** For a linear state with a fixed key, repeatedly applying Eq. (3) retains $\mathbf{k}_t^{(\ell)} = \mathbf{k}_t$, so all updates share the same right-hand direction and their sum remains Rank-1. Nonlinear memories such as TTT-MLP instead update their internal parameters across steps, allowing the effective direction $\mathbf{k}_t^{(\ell)}$ to evolve with $\ell$. At the level of a matrix-valued memory update, the accumulated multi-step correction can therefore be written as

$$\Delta \mathbf{S}_t^{[L]} = \sum_{l=1}^{L} \beta_t^{(l)} \boldsymbol{\delta}_t^{(l)} \mathbf{k}_t^{(l)\top}, \qquad \text{rank} \left( \Delta \mathbf{S}_t^{[L]} \right) \leq L. \tag{4}$$

Here, $\beta_t^{(l)}$ is the step coefficient, $\boldsymbol{\delta}_t^{(l)}$ is the gain-modulated residual correction, and $\mathbf{k}_t^{(l)}$ is the effective write direction at step $l$. When the corresponding correction vectors and directions are non-degenerate, the accumulated update can attain Rank-$L$. This motivates PRISM: rather than explicitly executing the state-dependent iterations, the following section uses learned input-conditioned quantities to retain the same multi-direction residual-refinement structure. Appendix B provides the detailed structural derivation.

### 3.3. The Rank-Parallel Trade-off

Modern hardware-efficient training relies on *parallel prefix scans* that require the recurrence to satisfy a *state-independence* condition:

**Definition 3.1 (Parallel Scan Compatibility).** A recurrence $\mathbf{S}_t = \mathbf{S}_{t-1} \mathbf{A}_t + \mathbf{B}_t$ admits parallel prefix scanning if and only if there exist input-only functions $f_A, f_B$ such that:

$$\mathbf{A}_t = f_A(\mathbf{x}_{\leq t}), \quad \mathbf{B}_t = f_B(\mathbf{x}_{\leq t}), \tag{5}$$

i.e., the operators $\mathbf{A}_t$ and $\mathbf{B}_t$ are independent of the hidden state $\mathbf{S}_{t-1}$.

The ideal multi-step solver (Eq. 4) fundamentally violates this condition: both the contextual gain $\sigma'(\mathbf{S}^{(l-1)} \mathbf{k}_t)$ and the residual $\mathbf{r}^{(l)} = \mathbf{v}_t - \sigma(\mathbf{S}^{(l-1)} \mathbf{k}_t)$ depend recursively on the hidden state. This creates a serial dependency chain with $O(NL)$ latency, precluding parallel computation.

**The Central Question.** This reveals a fundamental trade-off in existing methods (Table 1):

- **Linearize** the objective $\Rightarrow$ Rank-1 but parallel (Gated DeltaNet, PGDN).

- **Keep nonlinearity** and solve iteratively $\Rightarrow$ Rank-$L$ but serial (TTT).

Can we achieve **Rank-$L$ updates while preserving parallel scan compatibility**? In the next section, we show that this is possible through *amortization*: replacing the state-dependent terms with learned input-anchored proxies that emulate the multi-step trajectory without requiring sequential state access.

## 4. Method: The PRISM Architecture

We now present PRISM, which resolves the Rank-Parallel trade-off identified in §3.3. The key idea is to *amortize* the

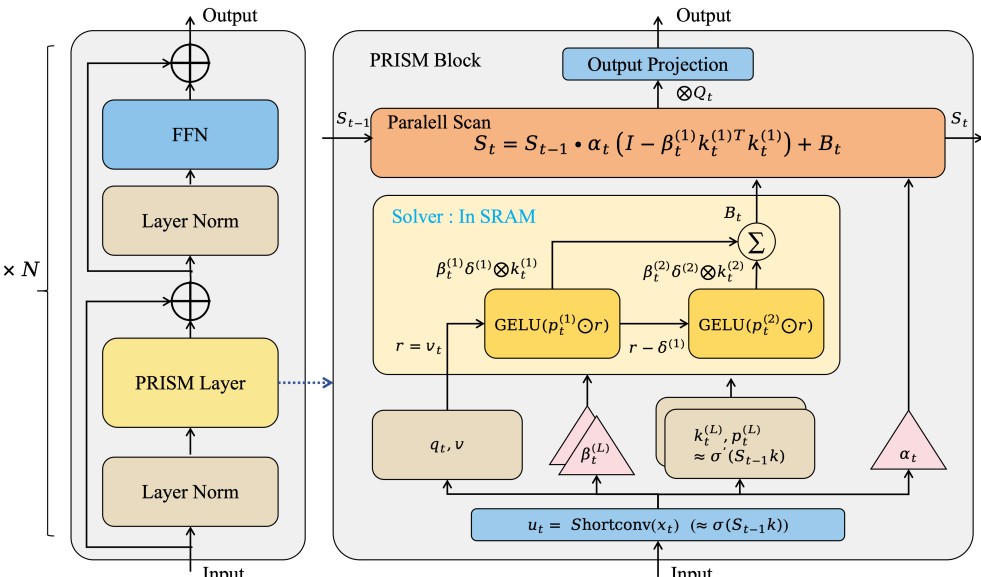

*Figure 1.* **The PRISM Architecture.** The framework resolves the Rank-Parallel trade-off through two mechanisms. **Phase 1 (Input-Anchored Proxy):** A ShortConv anchor computes a state-free proxy $\mathbf{u}_t \approx \mathbf{S}_{t-1}\mathbf{k}_t$. Learned predictors generate *Contextual Gain* vectors $\mathbf{p}^{(l)}$ (approximating $\phi'$) and *Basis Directions* $\mathbf{k}^{(l)}$ (approximating the multi-step gradient directions)—all from inputs alone. **Phase 2 (Rank-$L$ Accumulation):** An unrolled residual loop constructs the high-rank injection $\mathbf{B}_t = \sum_{l=1}^{L} \beta^{(l)} \boldsymbol{\delta}^{(l)} \mathbf{k}^{(l)\top}$. Each step performs greedy residual subtraction, expanding the update from Rank-1 to Rank-$L$. **State Update:** The accumulated Rank-$L$ injection is applied to a *decoupled linear recurrence* whose forgetting operator $\mathbf{A}_t$ is state-independent, preserving parallel scan compatibility.

multi-step solution of the ideal non-linear objective (Eq. 1) into a single parallel pass, by replacing state-dependent terms with learned input-anchored proxies.

## 4.1. Write-Forget Decoupling

Recall that any linear recurrence decomposes as $\mathbf{S}_t = \mathbf{S}_{t-1} \cdot \mathbf{A}_t + \mathbf{B}_t$, where $\mathbf{A}_t$ governs *forgetting* (which old information to decay) and $\mathbf{B}_t$ governs *writing* (what new information to inject). We ask: *which component benefits most from high-rank, non-linear expressivity?*

In Appendix E, we perform a spectral perturbation analysis and find a fundamental asymmetry:

- **Forgetting is robust.** Approximation errors in the multiplicative operator $\mathbf{A}_t$ accumulate sub-linearly—$O(\ln T)$ worst-case, $O(1)$ on average (Theorems E.2 & E.3). This means $\mathbf{A}_t$ can be safely approximated by a structured Rank-1 form without degrading long-term memory.

- **Writing is sensitive.** Additive errors in $\mathbf{B}_t$ accumulate linearly $O(T)$ in persistent memory channels. The system is hypersensitive to the rank-fidelity of the injection operator.

Based on this dichotomy, we **decouple** the two paths: the forgetting operator is fixed to the efficient Gated DeltaNet

form (right-multiplied as $\mathbf{S}_{t-1} \cdot \alpha_t(\mathbf{I} - \beta_t^{(1)} \mathbf{k}_t^{(1)} \mathbf{k}_t^{(1)\top})$), while the writing operator $\mathbf{B}_t$ is constructed as a Rank-$L$ injection via an iterative solver—all computed in parallel.

## 4.2. Input-Anchored Loop Unrolling

The ideal multi-step solver (Eq. 4) requires two state-dependent quantities at each step $l$: the *contextual gain* $\sigma'(\mathbf{S}^{(l-1)}\mathbf{k}_t)$ and the *residual* $\mathbf{r}^{(l)} = \mathbf{v}_t - \sigma(\mathbf{S}^{(l-1)}\mathbf{k}_t)$. Both depend on the hidden state $\mathbf{S}_{t-1}$, creating serial dependencies. We bypass this via a two-stage proxy mechanism.

**Stage 1: Anchor Construction.** We construct a causal input anchor using a short convolution:

$$\mathbf{u}_t = \text{ShortConv}(\mathbf{x}_t), \qquad \mathbf{u}_t \in \mathbb{R}^{d_u}. \qquad (6)$$

The anchor provides a local causal feature for predicting the refinement components. In Appendix F, we show that the approximation error decays exponentially with the effective memory window under fading-memory dynamics.

**Stage 2: Parallel Refinement Loop.** With the anchor $\mathbf{u}_t$ established, we unroll $L$ refinement steps—each producing an independent rank-1 component—without any inter-step state dependency.

**Initialization.** The residual is seeded by comparing the

target against the anchor $\mathbf{r}_t^{(1)} = \mathbf{v}_t$.

**Per-step computation.** For each step $l = 1, \ldots, L$, we compute three quantities from the anchor alone:

$$\mathbf{k}_t^{(l)} = \mathbf{W}_k^{(l)} \mathbf{u}_t \quad \text{(Direction: basis for step-} l \text{ update)}, \quad (7)$$

$$\mathbf{p}_t^{(l)} = \mathbf{W}_p^{(l)} \mathbf{u}_t \quad \text{(Gain: approximates } \phi'(\mathbf{S}^{(l-1)}\mathbf{k}_t)), \quad (8)$$

$$\beta_t^{(l)} = \mathbf{W}_\beta^{(i)} \mathbf{u}_t \quad \text{(Confidence: step-size gate)}, \quad (9)$$

where $\mathbf{W}_k^{(l)}, \mathbf{W}_p^{(l)}$ and $\mathbf{W}_\beta^{(l)} \in \mathbb{R}^{1 \times d}$ are trainable parameters. The update direction is formed by modulating the current residual with the simulated gain:

$$\boldsymbol{\delta}_t^{(l)} = \text{GELU}\left(\mathbf{p}_t^{(l)} \odot \mathbf{r}_t^{(l)}\right), \quad (10)$$

$$\mathbf{r}_t^{(l+1)} = \mathbf{r}_t^{(l)} - \boldsymbol{\delta}_t^{(l)}. \quad (11)$$

The greedy subtraction in Eq. 11 encourages subsequent steps to capture *complementary* error signals, enriching the information content of the total injection. The GELU activation in Eq. 10 provides the non-linear shaping that distinguishes PRISM from purely linear multi-rank approaches.

### 4.3. Rank-$L$ State Update

The $L$ rank-1 components are accumulated into the injection matrix:

$$\mathbf{B}_t = \sum_{l=1}^{L} \beta_t^{(l)} \cdot \boldsymbol{\delta}_t^{(l)} \mathbf{k}_t^{(l)^\top}. \quad (12)$$

Combined with the decoupled forgetting operator, the full PRISM state update is:

$$\mathbf{S}_t = \mathbf{S}_{t-1} \cdot \alpha_t \left(\mathbf{I} - \beta_t^{(1)} \mathbf{k}_t^{(1)} \mathbf{k}_t^{(1)^\top}\right) + \sum_{l=1}^{L} \beta_t^{(l)} \cdot \boldsymbol{\delta}_t^{(l)} \mathbf{k}_t^{(l)^\top}. \quad (13)$$

**Parallel Scan Compatibility.** In the form $\mathbf{S}_t = \mathbf{S}_{t-1}\mathbf{A}_t + \mathbf{B}_t$, both the transition matrix $\mathbf{A}_t = \alpha_t(\mathbf{I} - \beta_t^{(1)} \mathbf{k}_t^{(1)} \mathbf{k}_t^{(1)^\top})$ and the injection $\mathbf{B}_t = \sum_l \beta_t^{(l)} \boldsymbol{\delta}_t^{(l)} \mathbf{k}_t^{(l)^\top}$ are computed entirely from the input sequence (via the anchor $\mathbf{u}_t$) with no dependence on $\mathbf{S}_{t-1}$. Thus Eq. 13 satisfies Definition 3.1 and admits parallel prefix scanning.

**Computational Complexity.** The refinement loop adds $O(Ld)$ computation per token (for $L$ linear projections of the $d$-dimensional anchor), which is negligible compared to the $O(d^2)$ cost of the state multiplication. In practice, $L \in \{2, 4\}$ adds $< 5\%$ overhead relative to Gated DeltaNet, while delivering substantial quality gains.

## 5. Experiments

Our experiments validate PRISM's resolution of the Rank-Parallel trade-off along three axes:

- **RQ1 (Rank-$L$ Effectiveness):** Does PRISM's amortized Rank-$L$ injection match or exceed the modeling fidelity of Rank-1 baselines and explicit iterative solvers, while maintaining linear-time efficiency?

- **RQ2 (Parallel Scan Efficiency):** Does the Input-Anchored design preserve hardware-efficient parallelism as sequence length scales, achieving throughput comparable to Rank-1 models and orders of magnitude faster than serial solvers?

- **RQ3 (Mechanism Validation):** Do the individual components—iterative depth, non-linear activation, input anchor, and gain predictor—each contribute as theorized by our Write-Forget Decoupling and Rank Accumulation framework?

### 5.1. Experimental Setup

**Datasets.** We evaluate on four widely-used recommendation benchmarks: *Amazon Books*, *Amazon Movies*, *Amazon Electronics*, and *Yelp* (statistics in Appendix K). Sequential recommendation serves as a particularly stringent stress test for the Rank-1 bottleneck: user interests are inherently multi-modal, requiring the memory state to simultaneously encode multiple latent preference dimensions per interaction—precisely the scenario where Rank-1 updates are structurally insufficient.

**Baselines.** We organize baselines according to the Rank-Parallel taxonomy (Table 1):

1. **Rank-1, Parallel (One-Shot Linear):** SLA (Katharopoulos et al., 2020), GLA (Yang et al., 2023), GSA (Zhang et al., 2024), MoM (Du et al., 2025), Mamba-2 (Dao & Gu, 2024), Gated DeltaNet (Yang et al., 2024a)

2. **Rank-$L$, Serial (Multi-Step Explicit):** TTT (Sun et al., 2024), Titans (Behrouz et al., 2024), AT-LAS (Behrouz et al., 2025a)

3. **Full-Rank, Quadratic (Transformer):** SAS-Rec (Kang & McAuley, 2018), HSTU (Zhai et al., 2024)—upper bound at $O(N^2)$ cost.

PRISM occupies the previously empty cell: **Rank-$L$, Parallel**. Implementation details are in Appendix L.

**Metrics.** We report Hit@$K$ and NDCG@$K$ ($K \in \{100, 200, 500\}$) with per-dataset AUC for recommendation quality. For efficiency (RQ2), we measure training

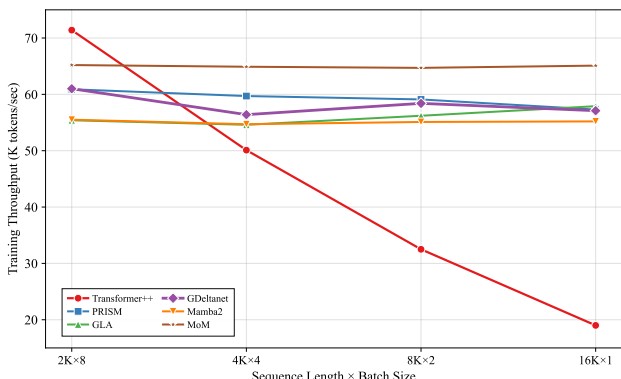

*Figure 2.* Training throughput (K tokens/s) of 0.13B models on an H20 GPU across sequence length × batch size configurations.

throughput (thousand tokens/s) on a single NVIDIA H20 GPU under identical batch configurations. All recurrent models use the materialization backend from Flash-Linear-Attention (Yang & Zhang, 2024).

### 5.2. Main Recommendation Performance (RQ1)

Table 2 (and Table 11 in Appendix M) summarize performance across four datasets.

**Rank-$L$ models dominate.** A clear trend emerges: models employing optimization-inspired or non-linear update mechanisms (Titans, PRISM, ATLAS) consistently outperform standard Rank-1 linear models (SLA, GLA, MoM). The top three linear models by Mean Rank all belong to this category, empirically validating that higher-rank injection captures complex interest evolution more effectively.

**Amortized $\approx$ Explicit.** PRISM achieves performance parity with—and in several cases surpasses—explicit serial solvers (TTT, Titans). On *Amazon Movies*, PRISM attains the highest AUC among all linear models, outperforming Titans. This confirms that Input-Anchored Loop Unrolling successfully distills the multi-step optimization trajectory into a single parallel pass, without requiring actual gradient descent at runtime.

**Narrowing the Transformer gap.** While $O(N^2)$ Transformers (SASRec, HSTU) retain a slight edge due to full context visibility, the gap is surprisingly narrow. On *Amazon Books*, PRISM's AUC is virtually identical to SASRec, suggesting that the "expressivity tax" of linearization becomes negligible with Rank-$L$ state tracking.

### 5.3. Training Efficiency (RQ2)

A core claim of PRISM is that Rank-$L$ expressivity need not sacrifice hardware efficiency. Figure 2 validates this by comparing training throughput as a function of sequence length.

**PRISM $\approx$ Rank-1 baselines.** PRISM maintains stable

throughput ($\sim$61K $\rightarrow$ 57K tokens/s) across all context lengths, performing on par with Gated DeltaNet, GLA, Mamba-2, and MoM (all within the 55–65K tokens/s range). The minor overhead ($<$16%) arises from the auxiliary GEMM operations for the $L$ direction/gain projections.

**Transformer degrades quadratically.** Transformer++ achieves the highest throughput at short sequences (71.4K tokens/s at 2K) thanks to highly optimized FlashAttention-2 (Dao, 2023), but suffers $3.8\times$ slowdown at 16K due to $O(N^2)$ complexity.

**TTT is $174\times$ slower.** TTT achieves only 0.34K tokens/s due to its inherently sequential state-dependent gradient computation. This starkly demonstrates the practical necessity of PRISM's Input-Anchored design: achieving Rank-$L$ updates *without* serial dependencies is not merely a theoretical nicety—it is a **$174\times$ throughput advantage** in practice.

### 5.4. Ablation Study: Deconstructing the Solver (RQ3)

To verify that each component of PRISM's solver contributes as hypothesized by our framework (§4), we conduct controlled ablations on *Amazon Electronics*.

**Iterative depth is the dominant factor.** Removing the iterative refinement ($L$=0, i.e., only a single solver step on top of the GDN base) causes the largest degradation, directly validating that multi-step refinement is critical for capturing complex user interest transitions.

**Non-linearity enables optimization-quality updates.** Removing GELU from the refinement loop ($\boldsymbol{\delta}^{(l)} = p^{(l)} \odot \mathbf{r}^{(l)}$ instead of $\mathrm{GELU}(p^{(l)} \odot \mathbf{r}^{(l)})$) costs $-0.9$pp AUC. This confirms that the non-linear shaping—which approximates the contextual gain $\sigma'(\mathbf{Sk})$ in the ideal solver—provides expressivity beyond what linear multi-rank injection alone can achieve.

**Anchor and gain provide grounding.** Removing Short-Conv or the gain predictor degrades global ranking, confirming that the optimization trajectory must be anchored in local context and modulated by an adaptive step size to be effective over long sequences.

### 5.5. Mechanistic Probing: Storage vs. Computation (RQ3)

Beyond aggregate metrics, we probe *why* PRISM outperforms Rank-1 baselines by disentangling memory capacity from computational expressivity. We design controlled tasks (Appendix N) in a resource-starved regime ($D$=16, $V$=64, $N$=128) to force architectural bottlenecks to surface.

Three findings emerge:

**(1) Memory capacity is not the bottleneck.** On pure associative retrieval (MQAR, Poly-Recall), all models achieve

*Table 2.* Recommendation performance (Hit@200, NDCG@200, AUC). Best linear results **bold**, second underlined.

| Model | Amazon_Books | | | Amazon_Movies | | | Amazon_Elec | | | Yelp | | | Mean Rank |
|---|---|---|---|---|---|---|---|---|---|---|---|---|---|
| | H@200 | N@200 | AUC | H@200 | N@200 | AUC | H@200 | N@200 | AUC | H@200 | N@200 | AUC | |
| **Rank-1, Parallel** | | | | | | | | | | | | | |
| SLA | 0.1129 | 0.0212 | 0.8866 | 0.1137 | 0.0227 | 0.7461 | 0.1290 | **0.0243** | 0.7023 | 0.1627 | 0.0311 | 0.9392 | 6.75 |
| GLA | 0.0879 | 0.0158 | 0.8752 | 0.1193 | 0.0222 | 0.7478 | 0.1196 | 0.0210 | 0.7008 | 0.1129 | 0.0220 | 0.8943 | 9.38 |
| MoM | 0.0854 | 0.0158 | 0.8705 | 0.1397 | 0.0281 | 0.7705 | 0.1333 | 0.0238 | 0.7042 | 0.1642 | 0.0306 | 0.9346 | 5.25 |
| GSA | 0.1226 | 0.0233 | 0.8870 | 0.1160 | 0.0222 | 0.7427 | 0.1318 | 0.0228 | 0.7087 | 0.1629 | 0.0307 | 0.9383 | 7.00 |
| Mamba-2 | 0.1234 | 0.0234 | 0.8872 | 0.1372 | 0.0276 | 0.7713 | 0.1338 | 0.0237 | 0.7157 | 0.1621 | 0.0306 | 0.9385 | 5.00 |
| GDeltaNet | 0.1214 | 0.0230 | 0.8844 | 0.1241 | 0.0253 | 0.7504 | 0.1333 | 0.0242 | **0.7159** | 0.1648 | 0.0308 | 0.9367 | 5.12 |
| **Rank-$L$, Serial** | | | | | | | | | | | | | |
| TTT | 0.1255 | 0.0234 | 0.8871 | 0.1288 | 0.0256 | 0.7591 | 0.1344 | 0.0234 | 0.6946 | 0.1636 | 0.0306 | 0.9375 | 5.25 |
| Titans | **0.1272** | **0.0243** | 0.8869 | 0.1358 | 0.0278 | 0.7652 | 0.1313 | 0.0233 | 0.7007 | **0.1653** | **0.0313** | **0.9395** | 3.62 |
| ATLAS | 0.1190 | 0.0223 | 0.8884 | 0.1367 | 0.0278 | 0.7710 | **0.1421** | **0.0243** | 0.7042 | 0.1629 | 0.0304 | 0.9383 | 5.00 |
| **Rank-$L$, Parallel (Ours)** | | | | | | | | | | | | | |
| PRISM | 0.1258 | 0.0238 | **0.8888** | **0.1411** | **0.0289** | **0.7727** | 0.1409 | 0.0237 | 0.7134 | 0.1637 | 0.0310 | 0.9393 | **2.62** |
| **Full-Rank, Quadratic** | | | | | | | | | | | | | |
| SASRec | 0.1138 | 0.0215 | 0.8910 | 0.1272 | 0.0244 | 0.7677 | 0.1438 | 0.0273 | 0.7293 | 0.1723 | 0.0325 | 0.9410 | – |
| HSTU | 0.1224 | 0.0233 | 0.8835 | 0.1399 | 0.0293 | 0.7748 | 0.1407 | 0.0250 | 0.7189 | 0.1595 | 0.0295 | 0.9324 | – |

*Table 3.* Component ablation on *Amazon Electronics*. Each row removes one component from the full PRISM model. All components contribute independently.

| Variant | Hit@K | | AUC |
|---|---|---|---|
| | 200 | 500 | |
| **PRISM (Full)** | **0.1409** | **0.2613** | **0.7134** |
| w/o Iterative Refinement ($L$=0) | 0.1155 | 0.2374 | 0.6805 |
| w/o Non-Linearity (GELU → Identity) | 0.1316 | 0.2477 | 0.7047 |
| w/o ShortConv Anchor | 0.1406 | 0.2584 | 0.7076 |
| w/o Gain Predictor ($p^{(l)}$) | 0.1383 | 0.2406 | 0.7098 |

*Table 4.* Mechanistic probing accuracy ($D$=16). **Logic** tasks reveal the Linearity Wall: Rank-1 models (LA, MoM) fail at chance level while PRISM matches the Transformer.

| Type | Task | TF | LA | MoM | PRISM |
|---|---|---|---|---|---|
| Memory | MQAR | **0.98** | **0.98** | **0.98** | **0.98** |
| | Poly-Recall | **1.00** | **1.00** | **1.00** | **1.00** |
| | Var. Tracking | **0.40** | 0.34 | 0.35 | 0.37 |
| Logic | Parity | **1.00** | 0.49 | 0.50 | **1.00** |
| | XOR | **1.00** | 0.50 | 0.49 | **1.00** |
| Structure | Mod. Add | 0.23 | 0.07 | 0.06 | **0.50** |
| | Palindrome | 0.49 | 0.49 | 0.50 | **0.99** |
| Control | MUX | **0.98** | **0.98** | 0.97 | **0.98** |
| | Silence Gate | **0.82** | 0.51 | 0.50 | 0.66 |

$\sim$100%. The state size ($d \times d$) is the binding constraint, not the update rule. PRISM's complex writing mechanism does not sacrifice storage efficiency.

**(2) The Linearity Wall is real.** On XOR and Parity, LA and MoM both collapse to chance, while PRISM achieves 100%. Crucially, MoM's failure proves that *spatial* rank expansion (multiple linear experts) cannot replace *computational* depth (iterative non-linear refinement). This directly validates PRISM's Rank-$L$ design: the benefit comes from the iterative non-linear *computation* within the injection operator, not merely from having more output directions.

**(3) Input-anchoring provides structural inductive bias.** On Palindrome and Modulo Addition, PRISM outperforms even the Transformer in this constrained regime. The Short-Conv anchor provides a strong prior for local structural patterns, whereas the Transformer must learn positional arithmetic from scratch with limited capacity.

## 6. Conclusion

In this work, we identify the **Rank-1 Bottleneck** as the primary expressivity limitation of efficient sequence models:

standard linear updates, while parallel, lack the optimization depth required to capture complex dependencies. We introduce **PRISM**, a framework that resolves this tension by achieving Rank-$L$ state updates while preserving parallel scan compatibility. Through **Write-Forget Decoupling**, we maintain stable linear decay while allocating capacity to a non-linear, Rank-$L$ injection operator. **Input-Anchored Loop Unrolling** bypasses the serial dependency of iterative solvers, collapsing a non-linear optimization loop into a single fused parallel operator. Our analysis proves that this design yields **Rank Accumulation** within a linear recurrence, expanding the hypothesis space beyond one-shot updates while maintaining the associative structure required for hardware-efficient prefix scans. Empirically, PRISM matches the fidelity of explicit serial solvers and narrows the gap with full-attention Transformers, while achieving **174$\times$** higher throughput than TTT, validating amortized optimization as a practical pathway beyond the Rank-1 barrier.

## Impact Statement

This paper presents work whose goal is to advance the field of efficient sequence modeling by bridging the gap between optimization fidelity and hardware-efficient parallel training. There are many potential societal consequences of our work, none (of) which we feel must be specifically highlighted here.

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

# A. Nomenclature and Definitions

To ensure clarity and consistency across the main text and appendices, we summarize the key mathematical symbols used in this paper in Table 5.

*Table 5.* Table of Notations.

| Symbol | Description |
|:---:|:---|
| $L$ | Number of refinement layers (solver steps). |
| $l$ | Index for refinement layers, $l \in \{1, \ldots, L\}$. |
| $\mathbf{S}_t \in \mathbb{R}^{d \times d}$ | Recurrent memory state at time step $t$. |
| $\mathbf{A}_t \in \mathbb{R}^{d \times d}$ | Forgetting operator (Linear Decay). |
| $\mathbf{B}_t \in \mathbb{R}^{d \times d}$ | Injection operator (Rank-$L$ Update). |
| $\mathbf{u}_t \in \mathbb{R}^d$ | Input-anchored proxy (from ShortConv). |
| $\mathbf{k}_t^{(l)} \in \mathbb{R}^d$ | Direction basis for step $l$ at time $t$. |
| $\mathbf{r}_t^{(l)} \in \mathbb{R}^d$ | Residual estimate for step $l$ at time $t$. |
| $\mathbf{p}_t^{(l)} \in \mathbb{R}^d$ | Contextual gain vector for step $l$ (approximates $\sigma'$). |
| $\boldsymbol{\delta}_t^{(l)} \in \mathbb{R}^d$ | Correction vector at step $l$. |
| $\beta_t^{(l)} \in \mathbb{R}$ | Confidence gate (step size) for step $l$. |
| $\alpha_t \in (0, 1]$ | Scalar decay gate. |

# B. From TTT-MLP to PRISM: A Structural Derivation

In this section, we trace the precise structural correspondence between TTT-MLP's implicit multi-step gradient descent and PRISM's explicit rank-$L$ injection, motivating PRISM as a "distillation" of TTT-MLP's computational structure onto a parallelizable linear state.

### B.1. TTT-MLP's Gate × Residual × Direction Structure

TTT-MLP (Sun et al., 2024) maintains an MLP $f(\mathbf{k}) = \mathbf{W}_2 \phi(\mathbf{W}_1 \mathbf{k})$ as its memory state, where $\mathbf{W}_2 \in \mathbb{R}^{d \times h}$ and $W_1 \in \mathbb{R}^{h \times d}$ are the weight matrices and $\phi$ is a nonlinear activation. At each token, TTT-MLP updates $\mathbf{W}_2$ via gradient descent on the reconstruction loss $\|\mathbf{v}_t - \mathbf{W}_2 \phi(\mathbf{W}_1 \mathbf{k}_t)\|^2$.

The $W_2$ gradient decomposes as:
$$\Delta \mathbf{W}_2 = \eta \cdot \mathbf{r}_t \cdot \phi(\mathbf{W}_1 \mathbf{k}_t)^\top, \tag{14}$$
where $\mathbf{r}_t = \mathbf{v}_t - \mathbf{W}_2 \phi(\mathbf{W}_1 \mathbf{k}_t)$ is the residual. Expanding column-wise:
$$\Delta \mathbf{W}_2[:, l] = \eta \cdot \underbrace{\phi(\mathbf{W}_l^\top \mathbf{k}_t)}_{\text{scalar gate}} \cdot \underbrace{\mathbf{r}_t}_{\text{residual}}, \tag{15}$$

where $\mathbf{w}_l$ is the $l$-th row of $\mathbf{W}_1$. Each hidden unit contributes a **scalar-gated copy of the residual**—this is the "gate × residual × direction" pattern.

### B.2. Multi-Step GD: Two Sources of Seriality

At gradient step $l$, the update becomes:
$$\Delta \mathbf{W}_2^{(l)} = \eta_l \cdot \mathbf{r}_t^{(l)} \cdot \phi(\mathbf{W}_1^{(l)} \mathbf{k}_t)^\top. \tag{16}$$

This creates **two serial dependencies**:

1. **Residual dependency**: $\mathbf{r}^{(l)} = \mathbf{v}_t - \mathbf{W}_2^{(l)} \phi(\mathbf{W}_1^{(l)} \mathbf{k}_t)$ depends on the updated $\mathbf{W}_2^{(l)}$ from the previous step.

2. **Direction dependency**: $\phi(\mathbf{W}_1^{(l)} \mathbf{k}_t)$ changes because $\mathbf{W}_1$ co-evolves with $\mathbf{W}_2$ during optimization.

Both dependencies force step-level seriality: step $l + 1$ cannot begin until step $l$ completes.

**Algorithm 1** PRISM Forward Pass: Input-Anchored Parallel Prediction

1: **Input:** Sequence $\mathbf{X} \in \mathbb{R}^{B \times N \times D}$, Parameters $\Theta$
2: // PHASE 1: INPUT-ANCHORED PROXY (FULLY PARALLEL)
3: $\mathbf{U} \leftarrow \text{SiLU}(\text{ShortConv}(\mathbf{X}))$ {Anchor: $\mathbf{u}_t \approx \mathbf{S}_{t-1}\mathbf{k}_t$}
4: $\mathbf{Q}, \mathbf{V} \leftarrow \text{Linear}(\mathbf{U})$
5: $\boldsymbol{\alpha} \leftarrow \sigma(\text{Linear}_\alpha(\mathbf{U}))$ {Decay gate}
6: *Compute per-step components for $l = 1 \ldots L$ (in parallel):*
7: **for** $l = 1$ **to** $L$ **in parallel do**
8: $\quad \mathbf{K}^{(l)} \leftarrow \text{Linear}_k^{(l)}(\mathbf{U})$ {Direction basis}
9: $\quad \mathbf{P}^{(l)} \leftarrow \text{Linear}_p^{(l)}(\mathbf{U})$ {Gain (approximates $\sigma'$)}
10: $\quad \boldsymbol{\beta}^{(l)} \leftarrow \text{Linear}_\beta^{(l)}(\mathbf{U})$ {Confidence gate}
11: **end for**
12: $\mathbf{R}^{(1)} \leftarrow \mathbf{V} - \mathbf{U}$ {Initial residual}
13: // PHASE 2: RANK-$L$ ACCUMULATION (FUSED INTO SCAN KERNEL)
14: **for** $t = 1$ **to** $N$ **do**
15: $\quad$ **Register:** $\mathbf{r}_t \leftarrow \mathbf{R}_t^{(1)}; \mathbf{B}_t \leftarrow \mathbf{0}$
16: $\quad$ **for** $l = 1$ **to** $L$ **do**
17: $\quad\quad \boldsymbol{\delta}_t^{(l)} \leftarrow \text{GELU}(p_t^{(l)} \odot \mathbf{r}_t)$ {Gain $\times$ Residual}
18: $\quad\quad \mathbf{B}_t \mathrel{+}= \beta_t^{(l)} \cdot \boldsymbol{\delta}_t^{(l)} \mathbf{k}_t^{(l)^\top}$ {Rank accumulation}
19: $\quad\quad \mathbf{r}_t \leftarrow \mathbf{r}_t - \boldsymbol{\delta}_t^{(l)}$ {Greedy subtraction}
20: $\quad$ **end for**
21: $\quad$ // STATE UPDATE (PARALLEL SCAN COMPATIBLE)
22: $\quad \mathbf{S}_t \leftarrow \mathbf{S}_{t-1} \cdot \alpha_t(\mathbf{I} - \beta_t^{(1)}\mathbf{k}_t^{(1)}\mathbf{k}_t^{(1)^\top}) + \mathbf{B}_t$
23: $\quad \mathbf{y}_t \leftarrow \text{OutProj}(\mathbf{S}_t \cdot \mathbf{q}_t)$
24: **end for**

## B.3. The Rank-1 Impossibility on Linear State

A natural question: can we achieve TTT-MLP's multi-direction capability on a *linear* state $\mathbf{S} \in \mathbb{R}^{d \times d}$? The gradient of the OCP objective with respect to $\mathbf{S}$ is:

$$\nabla_{\mathbf{S}}\mathcal{L} \propto \mathbf{r}_t \cdot \mathbf{k}_t^\top. \tag{17}$$

The column direction is locked to $\mathbf{k}_t^\top$. Performing $L$ steps of gradient descent with the *same* key accumulates to:

$$\sum_{l=1}^{L} \nabla^{(l)} = \left(\sum_{l=1}^{L} \mathbf{r}^{(l)}\right) \cdot \mathbf{k}_t^\top, \tag{18}$$

which remains **Rank-1** regardless of $L$. Multi-step gradient descent on a linear state with a single key *cannot* achieve multi-direction.

## B.4. Two Paths to Multi-Direction

- **TTT-MLP's path**: Use a nonlinear MLP state $\rightarrow \mathbf{W}_1$'s hidden layer implicitly provides $h$ directions $\phi(\mathbf{w}_l^\top \mathbf{k}_t) \rightarrow$ but weight updates create serial dependencies.

- **PRISM's path**: Keep the linear state $\mathbf{S} \rightarrow$ *explicitly* introduce $L$ learned directions $\mathbf{k}^{(l)} = \mathbf{W}_k^{(l)}\mathbf{u}_t$, precomputed from the anchor $\rightarrow$ Rank-$L$ without seriality.

## B.5. Structural Correspondence

PRISM reconstructs TTT-MLP's computational structure on a linear state:

$$\mathbf{B}_t = \sum_{l=1}^{L} \beta^{(l)} \left(\boldsymbol{\delta}^{(l)}\mathbf{k}^{(l)}\right). \tag{19}$$

*Table 6.* Structural correspondence between TTT-MLP and PRISM.

| TTT-MLP (implicit) | PRISM (explicit) |
|---|---|
| $\phi(\mathbf{W}_1^{(l)}\mathbf{k}_t)$ — direction from hidden layer | $\mathbf{k}^{(l)} = \mathbf{W}_k^{(l)}\mathbf{u}_t$ — learned projection from anchor |
| $\phi(\mathbf{W}_l^\top \mathbf{k}_t)$ — scalar gate | $p^{(l)} = \mathbf{W}_p^{(l)}\mathbf{u}_t$ — element-wise gate |
| $\mathbf{r}_t^{(l)}$ — decreases via $\mathbf{W}_2$ update (serial) | $\mathbf{r}_t^{(l+1)} = \mathbf{r}_t^{(l)} - \boldsymbol{\delta}^{(l)}$ — explicit subtraction (parallel) |
| Synchronized (non-parallelizable) | Decoupled (parallelizable, closed-form) |

### B.6. GDN as First-Step Special Case

Setting $L = 1$, $\beta^{(1)} = 1$, and noting that the initial residual $\boldsymbol{\delta}^{(1)} \approx \mathbf{v}_t - \mathbf{u}_t \approx \mathbf{v}_t$ (when the state is near-empty), we recover:

$$\mathbf{B}_t \approx \mathbf{v}_t \cdot \mathbf{k}_t^{(1)^\top}, \tag{20}$$

which structurally reduces to the Gated DeltaNet injection. PRISM with $L = 0$ (no solver steps) recovers the GDN update form.

## C. Analysis of Degeneracy Under Linear Mapping

This section formally analyzes the representational degeneracy inherent to linear attention models that employ linear projections for both key and value embeddings. We show that under this setup, the optimal state matrix admits a closed-form, sequence-independent solution, rendering online recurrent updates functionally redundant.

Consider a standard linear attention framework, where the key and value vectors for token $t$ are computed as linear transformations of the input token representation $\mathbf{x}_t$:

$$\mathbf{k}_t = \mathbf{W}_k \mathbf{x}_t, \quad \mathbf{v}_t = \mathbf{W}_v \mathbf{x}_t, \tag{21}$$

where $\mathbf{W}_k \in \mathbb{R}^{d \times d_e}$ and $\mathbf{W}_v \in \mathbb{R}^{d \times d_e}$ are learnable projection matrices. The model maintains a recurrent state matrix $\mathbf{S}_t$, updated online to minimize:

$$\mathcal{L}_t(\mathbf{S}_{t-1}) = \frac{1}{2} \|\mathbf{S}_{t-1}\mathbf{k}_t - \mathbf{v}_t\|_2^2. \tag{22}$$

Assume $\mathbf{W}_k$ is square and invertible ($d = d_e$, $\det(\mathbf{W}_k) \neq 0$). Setting the gradient to zero and substituting the linear definitions yields:

$$\mathbf{S}^\star = \mathbf{W}_v \mathbf{W}_k^{-1}. \tag{23}$$

This solution is both time-invariant and sequence-independent—the recurrent state fails to encode any sequential information, functioning merely as a fixed linear mapping. One way to overcome this degeneracy is to introduce nonlinearity $\sigma$ into the prediction model, leading to the OCP objective in Eq. 1.

## D. OCP Gradient and Parallelism Constraints

This section formalizes why the multi-step solution of the OCP objective (Eq. 1) is incompatible with parallel scan, complementing the main text derivation in §3.3.

### D.1. Parallelism Violation

Recall from §3 that the gradient of the OCP retrieval term with respect to $\mathbf{S}$ yields the update $\Delta \mathbf{S} = \beta_t(\mathbf{v}_t - \phi(\mathbf{S}\mathbf{k}_t)) \odot \phi'(\mathbf{S}\mathbf{k}_t) \cdot \mathbf{k}_t^\top$. In the $(\mathbf{A}_t, \mathbf{B}_t)$ formulation:

$$\mathbf{A}_t = \alpha_t \mathbf{I}, \quad \mathbf{B}_t = \beta_t(\mathbf{v}_t - \phi(\mathbf{S}_{t-1}\mathbf{k}_t)) \odot \phi'(\mathbf{S}_{t-1}\mathbf{k}_t) \cdot \mathbf{k}_t^\top. \tag{24}$$

Since $\mathbf{B}_t$ depends explicitly on $\mathbf{S}_{t-1}$ through both $\phi$ and $\phi'$, the composition $\theta_{t:t+1} = (\mathbf{A}_t \mathbf{A}_{t+1}, \ \mathbf{B}_t \mathbf{A}_{t+1} + \mathbf{B}_{t+1})$ cannot be pre-computed without materializing intermediate states, violating Definition 3.1 and forcing $O(N)$ sequential computation.

### D.2. PRISM's Resolution

PRISM replaces both state-dependent terms with input-anchored proxies:

- $\phi(\mathbf{S}_{t-1}\mathbf{k}_t) \approx \mathbf{u}_t = \text{ShortConv}(\mathbf{x}_t)$

- $\phi'(\mathbf{S}_{t-1}\mathbf{k}_t) \approx p_t^{(l)} = W_p^{(l)}\mathbf{u}_t$

This restores state-independence ($\mathbf{A}_t, \mathbf{B}_t$ depend only on inputs) while preserving the geometric structure of the multi-step update trajectory. The resulting operators satisfy Definition 3.1 and admit parallel prefix scanning.

## E. Sensitivity Analysis of Write-Forget Dynamics

### E.1. Theoretical Foundation: HiPPO, ODEs, and ZOH

Following HiPPO (Gu et al., 2020), the problem of online function approximation is cast as an ODE:

$$\dot{\mathbf{h}}(t) = \mathcal{A}(t)\mathbf{h}(t) + \mathcal{B}(t)\mathbf{x}(t). \tag{25}$$

Under Zero-Order Hold (ZOH) discretization, this yields $\mathbf{h}_k = \mathbf{A}_k\mathbf{h}_{k-1} + \mathbf{B}_k\mathbf{x}_k$ where $\mathbf{A}_k = \exp(\Delta\mathcal{A})$ inherits the spectral stability of $\mathcal{A}$.

### E.2. Spectral Bounds via Gated DeltaNet

**Lemma E.1** (Bounded Spectrum of the Forgetting Operator). *The forgetting operator $\mathbf{A}_t = I - \beta_t\mathbf{k}_t\mathbf{k}_t^\top$ with $\beta_t \in [0,1]$ and $\|\mathbf{k}_t\| \leq 1$ satisfies $\lambda(\mathbf{A}_t) \in [0,1]$.*

*Proof.* The matrix $\mathbf{A}_t = \mathbf{I} - \beta_t\mathbf{k}_t\mathbf{k}_t^\top$ has $d-1$ eigenvalues equal to 1 (for vectors $\perp \mathbf{k}_t$) and one eigenvalue $\lambda_{min} = 1 - \beta_t\|\mathbf{k}_t\|^2 \in [0,1]$. Thus $\rho(\mathbf{A}_t) \leq 1$, ensuring non-expansive dynamics. $\square$

**Remark.** Su (Su, 2026) independently proved a stronger result: the elements of the *inverse* of DeltaNet's core matrix $(I - \beta\mathbf{k}\mathbf{k}^\top)^{-1}$ are always bounded in $[-1, 1]$, providing additional numerical stability guarantees for the parallel scan computation. This complements our eigenvalue bound by showing that even the inverse (required during chunk-wise parallel computation) remains well-conditioned.

### E.3. Logarithmic Worst-Case Stability of Forgetting

**Theorem E.2** (Logarithmic Worst-Case Stability). *Under quadratic scaling of precision error ($\sigma^2 f(\lambda) \leq K\gamma^2$ where $\gamma = 1 - |\lambda|$), the worst-case accumulated error of the forgetting operator grows at most $O(\ln T)$.*

*Proof.* The per-lag variance is $V(\tau, \gamma) \leq K\tau\gamma^2 e^{-\gamma\tau}$. Maximizing over $\gamma$ yields $\gamma^* = 2/\tau$, giving the envelope $g(\tau) = 4Ke^{-2}/\tau$. Summing: $\sum_{\tau=1}^{T} g(\tau) \propto \sum_{\tau=1}^{T} 1/\tau \approx \ln T$. $\square$

**Theorem E.3** (Constant Average-Case Error). *For any fixed decay rate $\gamma$, the total accumulated error energy is $O(T)$, implying average error per step is $O(1)$.*

*Proof.* The lifetime error of a single injection is $\sum_{\tau=0}^{\infty} V(\tau, \gamma) \leq K\gamma^2 \cdot e^{-\gamma}/(1 - e^{-\gamma})^2 \approx K$ (for small $\gamma$). Summing over $T$ injections yields $O(T)$ total, hence $O(1)$ average. $\square$

### E.4. The Necessity of Rank-$L$ Injection

For injection errors (B-perturbation), setting $\gamma \to 0$ (persistent memory) trivially achieves $\sum_{\tau=1}^{T} \sigma_B^2 = T \cdot \sigma_B^2 = O(T)$.

**Summary:** A-Error (Forgetting) = $O(\ln T)$ worst-case, $O(1)$ average. B-Error (Writing) = $O(T)$ worst-case. This provides decisive justification for Write-Forget Decoupling: computational budget (Rank) must be prioritized for $\mathbf{B}$.

## F. Approximation Error of Input-Anchored Proxy

The true pre-activation is $\mathbf{z}_t = \mathbf{S}_{t-1}\mathbf{k}_t = (\sum_{i=0}^{t-1} \gamma^{t-i}\mathbf{v}_i\mathbf{k}_i^\top)\mathbf{k}_t$. The ShortConv proxy captures the window-$w$ terms: $\mathbf{u}_t = (\sum_{i=t-w}^{t-1} \gamma^{t-i}\mathbf{v}_i\mathbf{k}_i^\top)\mathbf{k}_t$. The approximation error is bounded by:

$$\|\mathbf{z}_t - \mathbf{u}_t\|_2 \leq \frac{\gamma^{w+1}}{1-\gamma}, \tag{26}$$

which decays exponentially with kernel size $w$.

## G. Rank Expansion via Multi-Component Injection

By sub-additivity of matrix rank:

$$\text{rank}(\mathbf{B}_t) = \text{rank}\left(\sum_{l=1}^{L} \beta_t^{(l)} \boldsymbol{\delta}_t^{(l)} \mathbf{k}_t^{(l)\top}\right) \leq L. \tag{27}$$

The bound is tight when $\{\mathbf{k}_t^{(l)}\}_{l=1}^L$ are linearly independent. Standard linear attention uses a single outer product, yielding Rank-1. PRISM expands this to a structured Rank-$L$ cone.

## H. Stability of Nested Non-Linear Refinement

**Lemma H.1** (Lipschitz Stability). *For GELU activation ($L_\phi \approx 1$) and bounded gain ($\|p^{(l)}\|_\infty \leq M$), a perturbation $\boldsymbol{\epsilon}$ in the residual produces output divergence:*

$$\|\boldsymbol{\delta}^{(l)} - \tilde{\boldsymbol{\delta}}^{(l)}\| \leq L_\phi \cdot M \cdot \|\boldsymbol{\epsilon}\|. \tag{28}$$

With $M \approx 1$ (due to LayerNorm) and $L \in \{2, 4\}$, the refinement loop remains numerically stable. The greedy subtraction $\mathbf{r} \leftarrow \mathbf{r} - \boldsymbol{\delta}$ further acts as negative feedback.

## I. Language Model Evaluation

To validate PRISM beyond sequential recommendation, we conduct language modeling experiments at the 130M parameter scale.

### I.1. Setup

**Training Data:** SlimPajama, 2B tokens. **Tokenizer:** Mistral (32K vocab).

**Model Configuration:** All models share identical hidden dimension, layers, and heads ($\sim$130M parameters), differing only in the recurrence core.

**Baselines:**

- **GDN** (Yang et al., 2024a): Rank-1 delta rule + gated decay (our $L = 0$ special case, no solver)

- **Mamba-2** (Dao & Gu, 2024): Structured state space model

- **EFLA** (Lei et al., 2025): Exact Flow Linear Attention

- **PGDN** (Tumma et al., 2026): Preconditioned GDN

- **PRISM**: Our method, $L = 2$ solver steps

### I.2. Perplexity Results

PRISM achieves the best perplexity on both validation and test sets (19.25 / 23.13), outperforming GDN by 0.15 PPL and Mamba-2 by 1.14 PPL. Notably, both EFLA and PGDN underperform GDN on this dataset, suggesting that exact flow and preconditioning provide less benefit than Rank-$L$ injection.

*Table 7.* Language modeling perplexity (SlimPajama 2B tokens, Mistral tokenizer).

| Model | Params | Best Val PPL ↓ | Final Test PPL ↓ |
|---|---|---|---|
| **PRISM** | 134.46M | **19.25** | **23.13** |
| GDN | 132.03M | 19.32 | 23.28 |
| EFLA | 132.03M | 19.41 | 23.33 |
| PGDN | 132.16M | 19.44 | 23.37 |
| Mamba-2 | 131.97M | 20.20 | 24.27 |

*Table 8.* Zero-shot evaluation across 9 benchmarks (SlimPajama 2B). Best in **bold**, second-best underlined. Avg ACC averages all 9 accuracy metrics.

| Model | Wiki↓ | LMB PPL↓ | LMB ACC↑ | PIQA↑ | Hella↑ | Wino↑ | ARC-e↑ | ARC-c↑ | BoolQ↑ | OBQA↑ | SciQ↑ | Avg↑ |
|---|---|---|---|---|---|---|---|---|---|---|---|---|
| **PRISM** | **34.68** | **27.00** | **19.8** | 58.2 | 26.4 | **51.3** | **35.4** | 19.6 | 59.4 | **27.2** | 73.4 | **40.1** |
| PGDN | 35.68 | 28.01 | 18.6 | 57.5 | 26.1 | 49.4 | **35.4** | 21.2 | **60.8** | 24.6 | 70.9 | 38.3 |
| EFLA | 35.51 | 28.50 | 18.6 | 58.4 | 26.1 | 50.5 | 34.6 | 20.8 | 54.2 | 26.4 | 73.2 | 38.1 |
| Mamba-2 | 37.35 | 30.26 | 17.9 | 58.3 | 25.8 | 50.3 | 34.1 | **21.8** | 49.8 | 25.4 | 70.0 | 37.1 |
| GDN | 35.19 | 28.82 | 18.8 | **58.9** | **26.7** | 49.3 | 33.3 | 20.9 | 46.9 | **27.2** | 70.1 | 36.9 |

## I.3. Zero-Shot Downstream Evaluation

PRISM achieves the highest average accuracy (40.1%), leading PGDN by 1.8pp and GDN by 3.2pp. The advantage is most pronounced on BoolQ (+12.5pp over GDN), WikiText PPL (34.68, best), and LAMBADA PPL (27.00, best), demonstrating that Rank-$L$ injection substantially enhances both perplexity and downstream task generalization.

## I.4. Ablation: Depth vs. Width Synergy

*Table 9.* Ablation (SlimPajama 2B). Training PPL differences are negligible, but evaluation metrics reveal large gaps—Rank-$L$'s true value lies in downstream generalization.

| Variant | Description | Wiki PPL ↓ | LMB PPL ↓ | Avg ACC ↑ | ΔACC |
|---|---|---|---|---|---|
| PRISM (full) | $L=2$, independent K, residual iter. | **34.68** | 27.00 | **40.1** | — |
| − Shared K | Solver steps share $\mathbf{k}^{(1)}$ | 34.69 | **26.02** | 39.8 | −0.3 |
| − Base K | Solver steps reuse GDN base key | 35.68 | 27.68 | 38.6 | −1.5 |
| − No residual | $r = v$ (no iterative subtraction) | 34.96 | 27.32 | 39.1 | −1.0 |
| − Single step ($L=1$) | GDN + 1-step solver | 35.26 | 32.55 | 37.2 | −2.9 |
| GDN baseline | Rank-1 | 35.19 | 28.82 | 36.9 | −3.2 |

**Key findings:**

- **Wiki PPL separates models clearly.** PRISM (34.68) > shared-K (34.69) > no-residual (34.96) > GDN (35.19) > $L=1$ (35.26). The full solver provides a consistent 0.5+ PPL improvement on external evaluation.

- **LAMBADA PPL reveals retrieval quality.** The single-step variant ($L=1$) suffers catastrophic degradation on LAMBADA (32.55 vs. 27.00 for full PRISM), demonstrating that Rank-$L$ is essential for precise long-range retrieval. This is the sharpest signal among all metrics.

- **Residual iteration matters for generalization.** Removing residual subtraction (−1.0pp ACC, LAMBADA 27.32) hurts more than sharing keys (−0.3pp ACC, LAMBADA 26.02), confirming that iterative refinement—not merely multi-direction projection—drives the generalization advantage.

- **All components contribute.** Every ablation degrades both evaluation PPL and Avg ACC, validating the synergistic design of PRISM's solver.

## J. Discussion

### J.1. Rank-$L$ Density vs. Rank-1 Sparsity

While standard linear models achieve $O(N)$ efficiency, their Rank-1 constraint means each token can only write along a single direction per step. PRISM overcomes this by *densifying* the injection operator to Rank-$L$ within the parallel scan framework. This enables multi-modal updates—simultaneously modifying orthogonal semantic subspaces (e.g., updating syntactic and semantic attributes independently) within a single timestep.

### J.2. The Memory Capacity Wall

Our Write-Forget Decoupling analysis reveals a deeper boundary: the forgetting operator is robust to linearization ($O(\ln T)$ error), implying complex non-linear gating yields diminishing returns for retention. Memory overwriting is inevitable in fixed-dimensional states ($\mathbf{S} \in \mathbb{R}^{d \times d}$), regardless of the writing algorithm. PRISM maximizes writing *fidelity* but does not expand the *container*. Approaches like Mixture-of-Memory (MoM) (Du et al., 2025) or GSA (Zhang et al., 2024) are complementary—PRISM serves as a high-density writing operator within expanded memory slots.

## K. Recommendation Dataset Descriptions

*Table 10.* Dataset statistics. Dense filtering (User interactions $\geq 40$) stress-tests long-range dependency modeling.

| Dataset | # Users | # Items | # Interactions | Avg. Length |
|---|---|---|---|---|
| Amazon_Books | 31,103 | 339,960 | 3,319,359 | 106.72 |
| Amazon_Movies | 9,429 | 58,636 | 1,142,976 | 121.22 |
| Amazon_Elecs | 1,869 | 33,135 | 123,147 | 65.89 |
| Yelp | 17,233 | 126,829 | 1,605,608 | 93.17 |

## L. Baseline Model Descriptions

We analyze all baselines through the generalized recurrence $\mathbf{S}_t = \mathbf{A}_t \mathbf{S}_{t-1} + \mathbf{B}_t$:

**Rank-1, Parallel (One-Shot Linear):**

- **SLA** (Katharopoulos et al., 2020): $\mathbf{A}_t = I$, $\mathbf{B}_t = \mathbf{v}_t \mathbf{k}_t^\top$. Pure accumulation.

- **GLA** (Yang et al., 2023): $\mathbf{A}_t = \mathrm{diag}(\alpha_t)$, $\mathbf{B}_t = \beta_t \mathbf{v}_t \mathbf{k}_t^\top$. Data-dependent diagonal decay.

- **Mamba-2** (Dao & Gu, 2024): $\mathbf{A}_t = \mathrm{diag}(\exp(-\Delta_t \mathbf{A}))$. Discretized input-dependent decay via SSD.

- **GSA** (Zhang et al., 2024): Slot-wise retention/input gates with complementarity ($\mathbf{f}_t \approx 1 - \mathbf{i}_t$).

- **MoM** (Du et al., 2025): Multiple independent GLA heads for capacity expansion.

- **Gated DeltaNet** (Yang et al., 2024a): $\mathbf{A}_t = \alpha_t(I - \beta_t \mathbf{k}_t \mathbf{k}_t^\top)$, $\mathbf{B}_t = \beta_t \mathbf{v}_t \mathbf{k}_t^\top$. One-step linear delta rule.

- **ATLAS** (Behrouz et al., 2025a): Enhanced delta rule with auxiliary mechanisms.

**Rank-$L$, Serial (Multi-Step Non-Linear):**

- **TTT** (Sun et al., 2024): $\mathbf{S}_t = \mathbf{S}_{t-1} - \eta \nabla_\mathbf{S} \ell(\mathrm{MLP}(\mathbf{x}_t; \mathbf{S}_{t-1}), \mathbf{y}_t)$. State-dependent gradient prevents parallel scan.

- **Titans** (Behrouz et al., 2024): Surprise-gated neural memory with momentum. Same serial constraint.

**Full-Rank, Quadratic (Transformer):**

- **SASRec** (Kang & McAuley, 2018): Self-attention with causal mask.

- **HSTU** (Zhai et al., 2024): Hierarchical sequential Transformer.

**PRISM** occupies the unique **Rank-$L$, Parallel** position: it computes $\mathbf{A}_t, \mathbf{B}_t$ independently of $\mathbf{S}_{t-1}$ (via input-anchored proxy) while achieving Rank-$L$ injection through iterative refinement.

## M. Additional Recommendation Results

*Table 11.* Recommendation performance (Hit@500, NDCG@500, AUC). Best linear results in **bold**, second-best underlined.

| Model | Amazon_Books | | | Amazon_Movies | | | Amazon_Elec | | | Yelp | | | Mean Rank |
|---|---|---|---|---|---|---|---|---|---|---|---|---|---|
| | H@500 | N@500 | AUC | H@500 | N@500 | AUC | H@500 | N@500 | AUC | H@500 | N@500 | AUC | |
| **Rank-1, Parallel** | | | | | | | | | | | | | |
| SLA | 0.2211 | 0.0342 | 0.8866 | 0.2111 | 0.0344 | 0.7461 | 0.2454 | 0.0382 | 0.7023 | 0.3219 | 0.0502 | 0.9392 | 5.88 |
| GLA | 0.1819 | 0.0270 | 0.8752 | 0.2122 | 0.0333 | 0.7478 | 0.2628 | 0.0381 | 0.7008 | 0.2186 | 0.0346 | 0.8943 | 7.62 |
| MoM | 0.1700 | 0.0259 | 0.8705 | 0.2322 | 0.0392 | 0.7705 | 0.2410 | 0.0367 | 0.7042 | 0.3106 | 0.0482 | 0.9346 | 8.25 |
| GSA | 0.2346 | 0.0367 | 0.8870 | 0.2090 | 0.0333 | 0.7427 | 0.2587 | 0.0380 | 0.7087 | 0.3164 | 0.0491 | 0.9383 | 6.25 |
| Mamba-2 | 0.2353 | 0.0368 | 0.8872 | 0.2388 | 0.0398 | 0.7713 | 0.2519 | 0.0378 | 0.7157 | 0.3200 | 0.0495 | 0.9385 | 4.25 |
| GDeltaNet | 0.2275 | 0.0357 | 0.8844 | 0.2212 | 0.0369 | 0.7504 | 0.2424 | 0.0371 | **0.7159** | 0.3163 | 0.0490 | 0.9367 | 7.00 |
| **Rank-$L$, Serial** | | | | | | | | | | | | | |
| TTT | **0.2398** | 0.0371 | 0.8871 | 0.2254 | 0.0372 | 0.7591 | 0.2403 | 0.0360 | 0.6946 | 0.3140 | 0.0486 | 0.9375 | 6.50 |
| Titans | 0.2362 | **0.0374** | 0.8869 | 0.2331 | 0.0394 | 0.7652 | 0.2628 | **0.0389** | 0.7007 | **0.3256** | **0.0505** | **0.9395** | **2.25** |
| ATLAS | 0.2330 | 0.0359 | 0.8884 | 0.2376 | 0.0399 | 0.7710 | **0.2629** | 0.0388 | 0.7042 | 0.3158 | 0.0487 | 0.9383 | 4.25 |
| **Rank-$L$, Parallel (Ours)** | | | | | | | | | | | | | |
| PRISM | 0.2383 | 0.0373 | **0.8888** | **0.2407** | **0.0409** | **0.7727** | 0.2613 | 0.0380 | 0.7134 | 0.3204 | 0.0497 | 0.9393 | 2.75 |
| **Full-Rank, Quadratic** | | | | | | | | | | | | | |
| SASRec | 0.2225 | 0.0345 | 0.8910 | 0.2281 | 0.0366 | 0.7677 | 0.2711 | 0.0425 | 0.7293 | 0.3279 | 0.0511 | 0.9410 | – |
| HSTU | 0.2310 | 0.0363 | 0.8835 | 0.2385 | 0.0411 | 0.7748 | 0.2574 | 0.0389 | 0.7189 | 0.3093 | 0.0475 | 0.9324 | – |

## N. Mechanistic Probing Details

To ensure reproducibility, we provide the detailed generation logic for the mechanistic probing tasks used in §5.5. Tasks are categorized into Memory Capacity, Non-Linear Logic, and Gating Control.

### N.1. Data Generation Logic

We employ a strict **Vocabulary Separation** strategy. The vocabulary $V$ is partitioned into disjoint sets: $\mathcal{V}_{data}$ for operands/values, $\mathcal{V}_{control}$ for triggers/queries, and $\mathcal{V}_{noise}$ for background noise. This prevents the model from relying on simple token-ID memorization.

### N.2. Task Instantiation Examples

Table 12 provides concrete input-output examples for all 9 tasks.

**Implementation Note.** For Logical tasks (Type II), we ensure that the relevant operands fall within the local receptive field of the ShortConv anchor. This enforces a test of the *update rule's expressivity* (can it approximate the non-linear function?) rather than long-range retrieval capacity.

### N.3. The Memory Capacity Wall

Our Write-Forget Decoupling analysis reveals a deeper boundary: the forgetting operator is robust to linearization ($O(\ln T)$ error), implying complex non-linear gating yields diminishing returns for retention. Memory overwriting is inevitable in fixed-dimensional states ($\mathbf{S} \in \mathbb{R}^{d \times d}$), regardless of the writing algorithm. PRISM maximizes writing *fidelity* but does not expand the *container*. Approaches like Mixture-of-Memory (MoM) (Du et al., 2025) or GSA (Zhang et al., 2024) are complementary—PRISM serves as a high-density writing operator within expanded memory slots.

---

**Algorithm 2** Comprehensive Synthetic Task Generation

---

**Input:** Seq Length $N$, Vocab $V$, Window $W$
**Init:** Fill sequence $X$ with noise $\sim \mathcal{V}_{noise}$
**// TYPE I: ASSOCIATIVE MEMORY (Storage)**
*Task 1: MQAR (Multi-Query Associative Recall)*
    Sample $K$ pairs $(k_i, v_i)$. Place at random positions. Query: $k_i$. Target: $v_i$.
*Task 2: Poly-Recall (Contextual Disambiguation)*
    Define contexts $C_1, C_2$. Map key $k$ to $v_1$ if $C_1$, $v_2$ if $C_2$. Query: $[C_{target}, k]$. Target: $v_{target}$.
*Task 3: Variable Tracking*
    Define chain: $a = val, b = a, c = b$. Query: $c$. Target: $val$.
**// TYPE II: NON-LINEAR LOGIC (Reasoning)**
*Task 4: Local XOR*
    Sample $a, b$. Label $= 1$ if $(a \bmod 2) \neq (b \bmod 2)$, else 0.
*Task 5: N-bit Parity*
    Sample $n$ bits. Label $= (\sum b_i) \bmod 2$.
*Task 6: Modulo Addition*
    Sample $a, b$. Label $= (a + b) \bmod M$.
*Task 7: Palindrome Detection*
    Sample $a, b, c$. Label $= 1$ if $a = c$, else 0.
**// TYPE III: GATING & CONTROL (Robustness)**
*Task 8: Silence Gate (Noise Filtering)*
    Sample trigger $T \in \{ON, OFF\}$. If $T = ON$: Target $= v$. If $T = OFF$: Target = NULL.
*Task 9: MUX Logic (Multiplexer)*
    Sample selector $S \in \{0, 1\}$, channels $ch_0, ch_1$. Label $= ch_S$.

---

*Table 12.* Examples of Mechanistic Probing Tasks ($D = 16, V = 64$). $\mathcal{N}$ denotes noise tokens. **Logic** tasks highlight the capability gap between Linear Attention and PRISM.

| Type | Task | Input Pattern | Underlying Logic | Target |
|------|------|---------------|------------------|--------|
| Memory (Capacity) | MQAR | `k=5,v=9 ... k=5` | Retrieval: $Mem[5] \rightarrow 9$ | 9 |
| | Poly-Recall | `CtxA,k=5,v=9 ... CtxA,k=5` | Contextual: $Mem[A][5] \rightarrow 9$ | 9 |
| | Var. Tracking | `a=7 ... b=a ... c=b ... c=?` | Pointer Chain: $c \rightarrow b \rightarrow a \rightarrow 7$ | 7 |
| Logic (Reasoning) | Local XOR | `[5, 8, XOR]` | $odd(5) \neq even(8) \rightarrow$ True | 1 |
| | N-bit Parity | `[1, 1, 1]` (N=3) | $1 + 1 + 1 = 3 \rightarrow odd$ | 1 |
| | Modulo Add | `[8, 4]` ($M = 10$) | $(8 + 4) \bmod 10 = 2$ | 2 |
| | Palindrome | `[4, 9, 4]` | $First(4) == Last(4) \rightarrow$ True | 1 |
| Control (Gating) | Silence Gate | `[OFF, k=5, v=9] ... k=5` | Gating: $Gate(OFF) \approx 0 \rightarrow$ Ignore | NULL |
| | MUX Logic | `[Sel=1, Ch0=3, Ch1=8]` | Routing: $Sel = 1 \rightarrow$ Pick $Ch1$ | 8 |

