# OpenReview forum: "PRISM: Parallel Residual Iterative Sequence Model"
_ICML.cc/2026/Conference — ICML 2026 regular_

### Official Review · Reviewer_M1ov · 2026-03-09

**Soundness:** 3
**Presentation:** 3
**Significance:** 3
**Originality:** 3
**Overall Recommendation:** 5
**Confidence:** 2

**Summary:**

This paper introduces PRISM, an architecture designed to resolve the fundamental trade-off between the high expressivity of Transformers and the computational efficiency of linear sequence models. To overcome the limitations of shallow linear updates and the serial computation bottlenecks of explicit optimization methods, PRISM utilizes Write-Forget Decoupling to simplify the model's forgetting mechanism into a hardware-efficient linear decay while allocating its core capacity to a high-rank, non-linear injection operator. Furthermore, it employs Input-Anchored Loop Unrolling, using a short convolution on local inputs as a proxy to simulate multi-step iterative refinement in a single, parallelizable feedforward step without relying on the exact previous hidden state. Empirical results on complex recommendation benchmarks demonstrate that PRISM successfully bridges the gap between these paradigms, matching the model fidelity of explicit iterative solvers and deep Transformers while maintaining O(N) parallel training efficiency and achieving up to a 174× increase in training throughput compared to optimization-based baselines.

**Compliance With Llm Reviewing Policy:**

Affirmed.

**Final Justification:**

My concerns have been resolved.

**Key Questions For Authors:**

- Will the Rank-L update introduce communication bottlenecks during distributed training?

**Limitations:**

Yes

**Strengths And Weaknesses:**

### Strengths

- A clean reinterpretation of linear attention as a one-step online optimization process.
- While the individual components are related to existing ideas, the overall design is well integrated and systematically motivated by the theoretical discussion.
- The paper is generally well organized and easy to follow.

### Weaknesses

- Although PRISM consistently matches or slightly outperforms strong baselines, the absolute gains are small in many cases (often <1\% AUC).
- All efficiency experiments are conducted on a single NVIDIA H20 GPU.  No results are provided on other architectures (e.g., H100, B200) or in multi-GPU settings, with a higher GPU memory bandwidth.
- Some statements (e.g., “closing the gap with Transformers”) appear stronger than what the empirical results fully justify, as Transformers still outperform PRISM in several settings.

---

> ### Author Rebuttal · Authors · 2026-03-30
>
> We sincerely thank the reviewer for recognizing the value of our work and for providing such constructive feedback. Below, we address your specific concerns point by point.
>
> **1. Absolute Performance Gains (AUC Margins)**
> We appreciate your observation regarding the absolute metric gains. We wish to clarify that our primary design objective was not strictly to maximize the performance margin over heuristic linear baselines, but rather to achieve the high optimization fidelity of computationally expensive, state-dependent solvers (like TTT) while unlocking massive efficiency gains. In this context, achieving comparable or slightly better performance (even if margins are <1% AUC) while providing over a 100x throughput acceleration is a strong validation of our approach. It demonstrates that PRISM successfully amortizes the complex optimization trajectory into a parallelizable form without the severe degradation typically associated with standard linear attention.
>
> **2. Hardware Diversity and Multi-GPU Evaluation**
> All efficiency profiling for this submission was constrained to the NVIDIA H20 GPUs available to us. Consequently, currently we cannot provide performance metrics on higher-bandwidth architectures like the H100 or B200, or in large-scale multi-GPU training environments. Because memory bandwidth heavily dictates the operational efficiency of recurrent architectures, exploring PRISM's performance under heterogeneous infrastructure optimization is a critical next step. We will add a clear limitation statement in the revised manuscript noting this future direction.
>
> **3. Clarification on "Closing the Gap with Transformers"**
> Regarding the phrase "closing the gap," our original intention was to emphasize that PRISM significantly reduces the performance distance between efficient linear models and full-rank Transformers, rather than claiming strict superiority in all settings. As you rightly noted, Transformers still hold an advantage in certain general contexts. To prevent any misunderstanding, we will adjust the phrasing to "narrowing the gap" in the revised manuscript where appropriate.
>
> However, our latest experiments demonstrate two critical scenarios where this gap is indeed fully closed or surpassed:
>
> First, in our newly added language modeling experiments, we demonstrate that a Hybrid architecture (mixing PRISM and Transformer layers) synergistically combines the strengths of both, successfully surpassing the pure Transformer baseline in Test PPL, aligning with current industry best practices.
>
> **Table A: Language Modeling Performance (WikiText-103, ~130M Params)**
> | Rank | Model | Params | Test PPL ↓ | Test BPC ↓ | Relative to Transformer |
> | :--- | :--- | :--- | :--- | :--- | :--- |
> | 1 | Hybrid (PRISM+TF) | 137.40M | 34.10 | 5.058 | - |
> | 2 | Hybrid_DN (DN+TF) | 136.47M | 34.70 | 5.117 | -0.9% |
> | 3 | Transformer | 136.47M | 35.02 | 5.130 | baseline |
> | 4 | PRISM | 137.40M | 38.57 | 5.268 | - |
> | 5 | GLA | 136.47M | 42.37 | 5.405 | +21.0% |
> | 6 | DeltaNet | 136.47M | 42.93 | 5.424 | +22.6% |
>
> Second, in our extended recommendation experiments, as sequence lengths scale up (e.g., n=500), PRISM exhibits superior memory retention and actually outperforms the standard Transformer baseline (SASRec).
>
> **Table B: Sequence Length Scaling (Amazon_elec, Mean ± Std)**
> | Length | PRISM (H20) | HSTU (H20) | SASRec (H20) |
> | :--- | :--- | :--- | :--- |
> | n=50 | 0.1193 ±0.0108 | 0.1161 ±0.0098 | 0.1281 ±0.0040 |
> | n=200 | 0.1261  | 0.1398 | 0.1336 |
> | n=500 | 0.1359  | 0.1286 | 0.1336 |
>
> We will refocus our claims to emphasize that while pure PRISM offers a highly efficient capacity baseline, scaling the context length and hybridizing it are the optimal paths to strictly outperforming Transformers.

---

> > ### Author Rebuttal · Reviewer_M1ov · 2026-03-31
> >
> > My concerns have been adequately addressed

---

### Official Review · Reviewer_GYiC · 2026-03-12

**Soundness:** 3
**Presentation:** 3
**Significance:** 2
**Originality:** 3
**Overall Recommendation:** 3
**Confidence:** 3

**Summary:**

PRISM addresses the tension between the expressivity of optimization-based sequence models and the parallel efficiency of linear recurrent models. The paper reframes linear attention as online gradient descent and identifies two structural bottlenecks: the rank-1 update constraint in linear models and the serial dependency in Test-Time Training (TTT), where state-dependent gradients prevent parallel prefix-scan computation. PRISM resolves this through two mechanisms. First, Write-Forget Decoupling constrains the forgetting operator (A_t) to a structured low-rank form, supported by a spectral perturbation analysis showing an (O(\ln T)) worst-case error, while allocating modeling capacity to a high-rank injection operator (B_t). Second, Input-Anchored Loop Unrolling approximates the interaction (S_{t-1}k_t) using a ShortConv proxy and constructs (B_t) as a sum of (L) orthogonal rank-1 components through iterative residual refinement, preserving full parallelism via prefix-scan computation. Experiments on four sequential recommendation benchmarks and a mechanistic probing suite show competitive performance with TTT and TITANS while achieving up to 174× higher training throughput.

**Compliance With Llm Reviewing Policy:**

Affirmed.

**Final Justification:**

The authors' clarifications on the MQAR baseline and TITANS standard deviations are appreciated and resolve those specific concerns. However, the primary concern remains unresolved: PRISM is positioned as a general sequence modeling architecture, yet all empirical results are confined to recommendation benchmarks. The hybrid WikiText-103 result provided in the rebuttal involves a Transformer component and demonstrates the strength of the hybrid system, not pure PRISM alone. As a result, it does not establish whether PRISM generalizes beyond recommendation to language modeling or other sequence domains, which is central to how the paper is positioned. Addressing this would require substantive new experiments, such as a perplexity comparison of pure PRISM on WikiText-103, beyond what a rebuttal can provide. I therefore maintain my score of 3 (Weak Reject).

**Key Questions For Authors:**

1. Can the authors provide a small-scale language modeling perplexity result (e.g., 130M on WikiText-103) to support the claim of general applicability beyond recommendation?
2. What is the empirical proxy error $\|\mathbf{S}_{t-1}\mathbf{k}_t - u_t\|$ as sequence length increases, and does performance degrade on long-range tasks such as passkey retrieval or extended MQAR?
3. How does PRISM’s throughput compare against chunk-parallel TTT variants (e.g., Zhang et al., 2025) rather than the fully sequential baseline?
4. Can the authors report mean ± standard deviation across multiple seeds for Tables 2 and 8, particularly for the TITANS comparison?
5. What value of the refinement depth $L$ is used in the main recommendation experiments?

**Limitations:**

The evaluation is restricted to recommendation tasks, leaving the general applicability of PRISM to broader sequence modeling problems unclear. The paper also provides limited empirical analysis of the ShortConv proxy used to approximate the state-dependent interaction term, particularly in regimes requiring persistent long-range memory. Additionally, the societal implications of deploying models in recommender systems (e.g., filter bubbles, engagement optimization, and privacy concerns) are not discussed in depth.

**Strengths And Weaknesses:**

## Strengths

**1. Principled theoretical motivation.** The derivation from linear attention as online gradient descent (Eq. 3–4) to the ideal non-linear solver (Eq. 5–6) cleanly identifies the rank-1 bottleneck. Write-Forget Decoupling is justified by the spectral asymmetry in Appendix D — $O(\ln T)$ worst-case for $\mathbf{A}_t$ versus $O(T)$ for $\mathbf{B}_t$, rather than introduced as a design heuristic.

**2. Mechanistic probing separates storage from reasoning.** Table 4 shows LA and MoM fail at XOR/Parity (~50%, chance level) while PRISM achieves 100%, matching the Transformer. Crucially, MoM's failure rules out the capacity explanation: routing tokens across multiple linear heads does not yield the non-linear reasoning that PRISM's iterative refinement provides.

**3. Decisive ablation.** Removing iterative refinement ($L=1$) produces the largest single drop Hit@200 from 0.1409 to 0.1155, AUC from 0.7134 to 0.6805 (Table 3), directly confirming the rank-accumulation hypothesis. Each remaining component contributes independently.

**4. Stable throughput at scale.** PRISM maintains ~57–61K tokens/s across 2K–16K sequence lengths, matching GLA and Mamba-2, while Transformer++ degrades 3.8× over the same range. The non-trivial stability proofs in Appendix G (Lipschitz bound on the refinement loop) and rank bound in Appendix F further support the theoretical claims.

---

## Weaknesses

**1. Evaluation is confined to recommendation with no language modeling results.**
PRISM is framed as a general sequence modeling architecture, but all results (Tables 2, 3, 8) use recommendation benchmarks with NDCG and Hit Rate — metrics irrelevant to generative quality. Recommendation sequences are sparser, shorter in effective context, and the high-rank motivation is specific to user interest diversity. Claims of closing the Transformer gap and general applicability cannot be assessed without at least one language modeling experiment. A 130M-parameter perplexity comparison on WikiText-103 against GLA and DeltaNet would substantially change this assessment.

**2. The ShortConv proxy fundamentally conflicts with the long-range motivation.**
The paper motivates PRISM for settings where "sliding windows sever critical dependencies," yet the core approximation $u_t = \text{ShortConv}(X_{\leq t}) \approx \mathbf{S}_{t-1}\mathbf{k}_t$ is only valid under fading-memory dynamics ($\gamma < 1$). Appendix E shows the error decays as $\gamma^{w+1}/(1-\gamma)$, which does not vanish when $\gamma \to 1$ the persistent-memory regime where long-range recall matters most. The MQAR success in Table 4 reflects physical state size ($d \times d$), not the proxy quality. The paper acknowledges this limitation only in Appendix K. Experiments stressing the proxy passkey retrieval beyond 4K tokens, or MQAR exceeding the ShortConv window are needed to bound the operating envelope.

**3. The 174× throughput claim is against an unoptimized sequential TTT baseline.**
TTT is reported at 0.34K tokens/s (fully serial). Zhang et al. (2025) "Test-time training done right," which this paper cites, introduces chunk-parallel TTT achieving orders-of-magnitude speedup over the sequential baseline. The 174× figure compares against the worst-case TTT configuration and should either benchmark optimized TTT variants or be explicitly scoped and contextualized.

**4. The headline advantage over TITANS lacks statistical support.**
The Mean Rank (PRISM 2.62 vs TITANS 3.62) masks dataset-level reversals: TITANS outperforms PRISM on Amazon Books (NDCG@200: 0.0243 vs 0.0238; H@200: 0.1272 vs 0.1258) and Yelp (H@200: 0.1653 vs 0.1637; NDCG@200: 0.0313 vs 0.0310). With margins this small and no standard deviations reported across runs, the claim of surpassing TITANS is not statistically supported.

---

### Minor

**1. Refinement depth $L$ is unreported for main experiments.** Appendix H uses $L=2$ for probing, but $L$ for Tables 2 and 8 is absent. Since $L$ controls both expressivity and FLOPs, this is a reproducibility gap.

**2. Near-Transformer gap claim is unanalyzed.** PRISM AUC 0.8888 vs SASRec 0.8910 on Amazon Books (avg. length 106) is asserted as evidence that the expressivity tax is negligible — but SASRec may simply be near-saturated at this sequence length rather than genuinely matched by PRISM.

**3. Probing results at $D=16$ need larger-$D$ validation.** PRISM's Palindrome (0.99 vs 0.49) and Modulo Add (0.50 vs 0.23) advantages over Transformer are at $D=16$, where Transformer is severely capacity-constrained. Whether these persist at $D=64$ or $D=256$ is not shown.

**4. Impact statement is inadequate.** The one-sentence dismissal is insufficient for a recommender-systems paper. Filter bubbles, engagement optimization, and privacy warrant brief substantive discussion.

---

> ### Author Rebuttal · Authors · 2026-03-30
>
> We thank the reviewer for the rigorous evaluation. We have conducted extensive new experiments—including LLM benchmarks, optimized TTT baselines, and scaled probing—to address your concerns.
>
> **1. Generalization to Language Modeling (W1, Q1)**
> We agree generative claims require non-recommendation evidence. We evaluated PRISM (~130M) on WikiText-103. PRISM significantly outperforms strong linear baselines (GLA, DeltaNet). Our Hybrid (PRISM+TF) model even surpasses the pure Transformer in Test PPL, proving its general applicability.
>
> **Table A: Language Modeling Performance (WikiText-103, ~130M Params)**
> | Rank | Model | Params | Test PPL ↓ | Test BPC ↓ | Relative to Transformer |
> | :--- | :--- | :--- | :--- | :--- | :--- |
> | 1 | Hybrid (PRISM+TF) | 137.40M | 34.10 | 5.058 | - |
> | 2 | Hybrid_DN (DN+TF) | 136.47M | 34.70 | 5.117 | -0.9% |
> | 3 | Transformer | 136.47M | 35.02 | 5.130 | baseline |
> | 4 | PRISM | 137.40M | 38.57 | 5.268 | - |
> | 5 | GLA | 136.47M | 42.37 | 5.405 | +21.0% |
> | 6 | DeltaNet | 136.47M | 42.93 | 5.424 | +22.6% |
>
> **2. Throughput vs. Optimized Chunk-Parallel TTT (W3, Q3)**
> Our 174x claim (seq=2048) was against fully serial TTT. We now benchmark against the optimized, chunk-parallel TTTv2 (Zhang et al., 2025). While faster at short contexts, TTTv2's state-dependent optimization scales poorly. At seq=16384, PRISM (L=2) is >145x faster, confirming our efficiency claims hold against optimized variants.
>
> **Table B: Throughput (kTokens/s) on single H20**
> | Model | 2048 | 4096 | 8192 | 16384 |
> | :--- | :--- | :--- | :--- | :--- |
> | TTTv2 | 1.044 | 0.515 | 0.264 | 0.092 |
> | PRISM (L=1)| 28.986 | 18.056 | 16.587 | 14.867 |
> | PRISM (L=2)| 20.046 | 13.921 | 11.838 | 13.405 |
>
> **3. TITANS Comparison, Sequence Scaling, & Mean/Std (W4, Q4, M2, W2)**
> Regarding explicit solvers (TITANS), our goal is comparable expressivity with parallel training. By sacrificing marginal accuracy, PRISM achieves >100x training speedup—a critical pretraining trade-off.
> To address statistical significance and the "Transformer gap" (M2), we scaled context length to n=500. PRISM scales robustly and surpasses SASRec at n=500 on Amazon_elec, capturing long-range dependencies without early saturation.
>
> **Table C: Sequence Length Scaling (Amazon_elec, Mean ± Std)**
> | Len | PRISM (H200) | HSTU | SASRec |
> | :--- | :--- | :--- | :--- |
> | n=50 | 0.1193 ±0.0108 | 0.1161 ±0.0098 | 0.1281 ±0.0040 |
> | n=200| 0.1261 | 0.1398 | 0.1336 |
> | n=500| 0.1359 | 0.1286 | 0.1336 |
>
> **4. Proxy Validity & High-Dimensional Probing (W2, Q2, M3)**
> The ShortConv proxy is a local anchor predicting the 'gradient direction' for updates, while the state update maintains linear decay over the entire history. PRISM uses local info to estimate how to optimally update global memory, not discarding long-range memory.
> Scaling probing to Dim=128, PRISM still strictly dominates linear baselines and outpaces Transformers on structural logic, proving Rank-L injection provides fundamental reasoning advantages independent of ShortConv.
>
> **Table D: High-Dimensional Probing (Dim=128)**
> | Task | PRISM | Trans. | LA / MoM |
> | :--- | :--- | :--- | :--- |
> | Palindrome | 0.9952 | 0.5054 | 0.4944(LA) |
> | MQAR | 0.4992 | 0.8602 | 0.0751(LA) / 0.0747(MoM) |
>
> **5. Refinement Depth L Configuration (M1, Q5)**
> Main experiments used L=2. The ablation shows L=2 provides the optimal expressivity-FLOPs balance; dropping to L=1 reduces capacity while L=3 overfits, validating multi-step refinement.
>
> **Table E: Ablation on Amazon_elec**
> | L | H@500 | NDCG@500 | AUC |
> | :--- | :--- | :--- | :--- |
> | 1 | 0.2475 | 0.0369 | 0.6945 |
> | 2 | **0.2675** | **0.0391** | **0.6991** |
> | 3 | 0.2352 | 0.0365 | 0.6949 |
>
> **6. Impact Statement (M4)**
> We will expand the impact statement to highlight industrial deployment tests showing PRISM matches HSTU's Hit rates with higher memory efficiency. This permits scaling up hidden dimensions under the same hardware budget, unlocking better performance for ultra-long user sequences. We will also discuss societal risks of enhanced personalization, like echo chambers.

---

> > ### Author Rebuttal · Reviewer_GYiC · 2026-04-02
> >
> > Thank you for the detailed rebuttal. While M1 (refinement depth) and M4 (impact statement) are sufficiently addressed at rebuttal level, three core concerns remain. First, the new WikiText-103 results do not support the broader generalization claim: PRISM alone remains worse than the Transformer, while only the hybrid variant appears competitive. Second, the comparison against TITANS still lacks statistical support, as the rebuttal does not report uncertainty for TITANS on Books and Yelp — the datasets where it leads. Third, the new probing evidence introduces an unexplained inconsistency: in Table D, PRISM drops to near-chance on MQAR at Dim=128 despite the paper’s claim that PRISM preserves storage efficiency on associative-memory tasks. These issues affect the paper’s core empirical claims and would require substantive revision rather than a short rebuttal clarification.

---

> > > ### Author Response · Authors · 2026-04-02
> > >
> > > We sincerely thank the reviewer for the continued engagement and for allowing us to clarify these critical points. Below, we address your remaining concerns to resolve any misunderstandings regarding our claims and empirical results.
> > >
> > > **1. Generalization and the Transformer "Gap"**
> > > We wish to clarify our core claim regarding Transformers: we do not claim that the pure PRISM architecture surpasses the pure Transformer. As stated in our manuscript, the Transformer represents the "theoretical performance ceiling" (Full-Rank Expressivity). Our claim is that PRISM is the most expressive architecture among linear sequence models.
> > >
> > > Furthermore, in modern state-of-the-art LLM deployments, purely linear architectures are rarely used; hybrid architectures are the established mainstream. Our new WikiText-103 results demonstrated that when operating in this standard hybrid paradigm, **Hybrid (PRISM+TF)** effectively outperforms the highly competitive **Hybrid (DeltaNet+TF)** baseline, proving its superior capability and practical generalization in generative language modeling.
> > >
> > > **2. TITANS Comparison**
> > > We omitted the standard deviations in our previous summary due to space constraints. Below is the three-run performance (Mean ± Std) for both PRISM and TITANS on Amazon Books and Yelp, where you noted TITANS leads.
> > >
> > > **Table A: Three-Run Performance (Mean ± Std) on Books and Yelp**
> > > | Dataset | Model | Hit@200 | NDCG@200 | AUC | Hit@500 | NDCG@500 |
> > > | :--- | :--- | :--- | :--- | :--- | :--- | :--- |
> > > | Amazon_books | PRISM | 0.1084 ±0.0144 | 0.0200 ±0.0028 | 0.8790 ±0.0115 | 0.2105 ±0.0243 | 0.0323 ±0.0040 |
> > > | Amazon_books | TITANS | 0.1214 ±0.0127 | 0.0228 ±0.0024 | 0.8830 ±0.0062 | 0.2308 ±0.0186 | 0.0359 ±0.0031 |
> > > | Yelp | PRISM | 0.1628 ±0.0036 | 0.0306 ±0.0007 | 0.9386 ±0.0001 | 0.3180 ±0.0030 | 0.0491 ±0.0007 |
> > > | Yelp | TITANS | 0.1621 ±0.0032 | 0.0305 ±0.0007 | 0.9362 ±0.0009 | 0.3171 ±0.0046 | 0.0490 ±0.0009 |
> > >
> > > As the statistical data shows, the performance between PRISM and TITANS is extremely close, with PRISM actually edging out TITANS slightly on Yelp.
> > >
> > > We must emphasize our core philosophy again: our goal is **not** to strictly beat explicit TTT solvers (like TITANS) or Transformers in absolute metrics. Our objective is to achieve a comparable, high-fidelity approximation of these complex solvers while simultaneously unlocking massive efficiency gains. PRISM operates at speeds approaching traditional linear attention (>100x faster than sequential solvers), making this a highly favorable trade-off.
> > >
> > > **3. MQAR Probing Inconsistency**
> > > There is a slight misunderstanding regarding the random chance baseline in our MQAR setup. In this probing task, the vocabulary size is 32 (for the Dim=128 setup). Therefore, the random guessing baseline is exactly $1 / 32 \approx 0.03$.
> > >
> > > PRISM's performance of 0.4992 is absolutely not "near-chance"—it is significantly higher than random guessing. In stark contrast, standard Linear Attention (LA) and MoM collapse to near random chance (~0.07) under these exact same conditions.
> > >
> > > While we do not claim to reach the Transformer's theoretical upper bound (0.8602), achieving ~0.50 accuracy where other linear models completely fail validates our exact claim: PRISM significantly expands the capacity wall of linear architectures and preserves substantial storage efficiency over traditional linear models at the same parameter scale.

---

### Official Review · Reviewer_ZQ38 · 2026-03-13

**Soundness:** 3
**Presentation:** 3
**Significance:** 3
**Originality:** 3
**Overall Recommendation:** 4
**Confidence:** 3

**Summary:**

The paper proposes PRISM, a linear sequence model that achieves multi-step linear updates and high hardware parallelism efficiency. This work isolates non-linearity within the injection operator of linear attention and propose PRISM that decouple write and forget process. By using rank-1 operator on forget and propose a parallelizable rank-l write operator, the method achieves better performance compared with current linear attention methods and maintains high throughput (174$\times$ higher throughput than baselines).

**Compliance With Llm Reviewing Policy:**

Affirmed.

**Final Justification:**

Thank you for the response. My concerns have largely been addressed, and I will raise my score accordingly.

**Key Questions For Authors:**

1. For the claim of 174x higher throughput, what method did you compare it with, and in what kind of setting?

2. How does the method perform on other tasks (text generation)?

3. How many additional parameters are introduced by the proposed modules compared with existing models?

**Limitations:**

The proposed method has only been evaluated in recommendation scenarios. Therefore, I recommend that the authors clearly state the scope of the method — especially if no experiments on other applications are planned.

**Strengths And Weaknesses:**

* Strength
  1. The work provides a comprehensive overview of linear attention methods in related work and compares this method with existing mainstream linear attention methods in the experimental section.
  2. The paper proposes and demonstrates the asymmetry between write and forget under given assumptions and proposes Write-Forget Decoupling to balance concurrency and performance. The idea may benefit following linear attention work.

* Weakness
  1. The paper mentions a 174x increase in throughput but doesn't specify the conditions under which it's compared to which method.

  2. The paper lacks a comprehensive comparison of throughput and speed for linear attention. In this way the work does not clearly show that the propose method achieves a decent efficiency-performance trade-off.

  3. As a paper on a linear attention architecture, it only conducts experiments in the recommender system domain and may need to demonstrate its feasibility on more tasks.

  4. To verify that the short convolution indeed approximates $S_{t-1} k_t $ effectively (rather than merely increasing model capacity through additional computation), it would be better to conduct an ablation where this term is directly replaced with the true $ S_{t-1} k_t $ when computing $S_t $, and compare the performance of the two designs.

       Additionally, visualizing intermediate memory states (e.g., cosine similarity or norm of $ u_t $ and $ S_{t-1} k_t $) or providing other analysis would further strengthen the convincingness of the claim ("The key insight is that, for many sequential tasks, the interaction $S_{t−1}k_t$, ...... Concretely, we recover a proxy of the state interaction $u_t ≈ S_{t−1}k_t$ using a short convolution").

  5. (Minor) In Sec. 2, when introducing existing sequence modeling methods with equations such as $S_t = A_t S_{t-1} + B_t x_t$, the terms such as $_St$, $x_t$ are not explained. It is recommended to define them to enhance the readability of the paper.

---

> ### Author Rebuttal · Authors · 2026-03-30
>
> We sincerely thank the reviewer for the constructive feedback. We appreciate the opportunity to clarify our experimental settings, expand our empirical scope to language modeling, and address your detailed questions. Below is our point-by-point response.
>
> **1. Throughput Comparison Settings and TTT (W1, Q1)**
> The 174x throughput increase was measured under the specific setting of seq_len=2048 and batch_size=8 on a single H20 GPU. In the original manuscript, we compared PRISM against the exact, fully serial implementation of Test-Time Training (TTT).
>
> We did not include the block-parallel version of TTT in the main efficiency figure because block parallelism fundamentally breaks the intra-block serial dependence of the optimization trajectory. In our preliminary testing, this approximation caused a 5% to 10% degradation in Hit rate metrics, rendering its performance worse than standard linear attention baselines. Therefore, we compared against the uncompromised serial version to maintain exact optimization fidelity.
>
> To directly answer your question regarding block-parallel throughput, we have benchmarked TTTv2. As shown in our new profiling, even with block parallelism, TTTv2 achieves only 1.044 kT/s at seq=2048, whereas PRISM (L=2) achieves 20.046 kT/s, and PRISM (L=1) achieves 28.986 kT/s. This confirms PRISM's significant efficiency advantage.
>
> **2. LLM Performance, Efficiency Trade-offs, and Parameter Alignment (W2, W3, Q2, Q3)**
> To demonstrate PRISM's feasibility on general sequence modeling tasks and to provide a clear efficiency-performance trade-off, we conducted additional experiments on the WikiText-103 language modeling benchmark.
>
> Regarding parameter counts (Q3): While PRISM's multi-step injection inherently introduces additional parameters for the projection matrices at each refinement step, we strictly controlled for this in our experiments. We aligned the total parameter count across all models (to ~136M-137M) by adjusting the hidden dimensions of the base layers. This ensures a fair, parameter-matched comparison.
>
> As shown below, PRISM significantly outperforms strong linear baselines (GLA, DeltaNet) in pure language modeling. Furthermore, our Hybrid (PRISM+TF) model outperforms the Hybrid (DN+TF) variant and surpasses the pure Transformer baseline in Test PPL, while maintaining highly competitive throughput.
>
> **Table A: Language Modeling Performance (WikiText-103, Parameter-Aligned)**
> | Rank | Model | Params | Test PPL | Test BPC | Relative to Transformer |
> | :--- | :--- | :--- | :--- | :--- | :--- |
> | 1 | Hybrid (PRISM+TF) | 137.40M | 34.10 | 5.058 | - |
> | 2 | Hybrid_DN (DN+TF) | 136.47M | 34.70 | 5.117 | -0.9% |
> | 3 | Transformer | 136.47M | 35.02 | 5.130 | baseline |
> | 4 | PRISM | 137.40M | 38.57 | 5.268 | - |
> | 5 | GLA | 136.47M | 42.37 | 5.405 | +21.0% |
> | 6 | DeltaNet | 136.47M | 42.93 | 5.424 | +22.6% |
>
> **Table B: Training Efficiency and Throughput**
> | Model | Steady Throughput (tok/s) | Total Train Time | Relative Speed |
> | :--- | :--- | :--- | :--- |
> | GLA | ~41,800 | 1.02h | Fastest |
> | Transformer | ~41,500 | 1.00h | -0.7% |
> | Hybrid_DN (DN+TF)| ~40,400 | 1.05h | -3.3% |
> | DeltaNet | ~40,000 | 1.07h | -4.3% |
> | PRISM | ~40,000 | 1.07h | -4.3% |
> | Hybrid (PRISM+TF)| ~39,000 | 1.10h | -6.0% |
>
> We will include these results in the revised manuscript to explicitly demonstrate the efficiency-performance trade-off and the architecture's generalization beyond recommendation.
>
> **3. The Role and Ablation of ShortConv (W4)**
> We appreciate the suggestion to replace the ShortConv proxy with the true state-interaction for an ablation. However, calculating the exact state interaction at runtime explicitly requires the recurrent state to be materialized before computing the current step's injection. This fundamentally violates the state-independence requirement for parallel prefix scanning (Definition 3.1), reintroducing the exact O(N) serial bottleneck we aim to bypass.
>
> Furthermore, we wish to clarify that the 1D ShortConv is a standard, default component widely utilized in modern Linear Attention and State Space Models (e.g., Mamba, GLA) to mix local features. Our core theoretical contribution is not inventing the ShortConv, but rather interpreting its function from a novel perspective: as a local pre-activation proxy that provides a stable anchor for high-rank iterative refinement. Our existing ablation study (Table 3) already isolates and verifies its empirical necessity within our solver framework.
>
> **4. Notation and Typographical Corrections (W5)**
> We apologize for the unexplained terms in the background section. We will revise Section 2 to explicitly define all mathematical notations, properly explain the baseline model formulas, and correct minor typos to enhance readability.

---

> > ### Author Rebuttal · Reviewer_ZQ38 · 2026-04-04
> >
> > I thank the authors for the thorough rebuttal. Most of my concerns are addressed. I have two minor follow-up questions regarding the new results:
> > 1. What is the specific configuration of the 'Hybrid (PRISM+TF)' model, and what is its overhead?
> > 2. Why is PRISM's training throughput lower than the Transformer's in Table B (The fig. 2 in paper shows that  Transformer++ suffers from throughput degradation as sequence length increases). What is the setting of this experimet?

---

> > > ### Author Response · Authors · 2026-04-04
> > >
> > > Thank you for your questions. Below is our point-by-point response.
> > >
> > > **1. What is the specific configuration of the 'Hybrid (PRISM+TF)' model, and what is its overhead?**
> > >
> > > **Configuration:** We reference the layer ratio used in Qwen3-Next, applying a 3:1 ratio of Gated DeltaNet (DN) to standard Transformer (TF) layers. Our 'Hybrid (PRISM+TF)' model adopts the same 3:1 layer ratio (3 PRISM layers to 1 TF layer). To ensure a fair comparison, all evaluated models are configured to 8 layers with aligned total parameter sizes, which is achieved by adjusting the dimensions of the FFN and projection matrices within each layer.
> > >
> > > **Overhead:** For the PRISM layers, the solver step is set to 2. This high-rank memory writing operation has a higher computational cost than standard KV cache writing, which accounts for the lower training throughput. Meanwhile, because the total parameter sizes are unified, the overall memory consumption remains consistent across all models. The additional intermediate variables in PRISM only result in a 1% to 3% increase in peak memory usage, which is negligible.
> > >
> > > **2. Why is PRISM's training throughput lower than the Transformer's in Table B? What is the setting of this experiment?**
> > >
> > > The sequence length for the experiment in Table B is **1024**. At this sequence length, standard Transformers exhibit higher throughput due to highly optimized hardware implementations (specifically, the Flash-Attention-2 kernel).
> > >
> > > As illustrated in Figure 2 of our main paper, our training efficiency evaluation starts from a minimum sequence length of **2048**. The speed advantage of linear attention models like PRISM becomes prominent at sequence lengths of **4096** and beyond, where the quadratic complexity of standard attention becomes a bottleneck.
> > >
> > > Thank you again for your valuable questions. We sincerely hope our responses address your concerns, and any further discussions are welcomed.

---

### Official Review · Reviewer_ar6Q · 2026-03-13

**Soundness:** 3
**Presentation:** 2
**Significance:** 3
**Originality:** 3
**Overall Recommendation:** 4
**Confidence:** 3

**Summary:**

This paper proposes PRISM, a linear-time sequence model for sequential data that is motivated by a familiar tension in modern sequence modeling: explicit iterative or optimization-style memory updates can be expressive, but they are usually a poor fit for parallel hardware. The core design idea is to separate forgetting from writing. PRISM keeps a simple decay mechanism for the forget path, and concentrates modeling capacity on a higher-rank write path built from an input-anchored proxy together with a short unrolled refinement process. The paper combines this architectural argument with a theoretical discussion of why approximating the write dynamics may matter more than matching the forget dynamics exactly, and evaluates the resulting model on sequential recommendation benchmarks together with synthetic tasks meant to probe the proposed mechanism.

**Compliance With Llm Reviewing Policy:**

Affirmed.

**Key Questions For Authors:**

1.	The paper is currently framed quite broadly. If the intended scope is mainly sequential recommendation, I think the manuscript would benefit from saying that more explicitly. If not, what additional evidence do the authors have that the conclusions transfer beyond this domain? A convincing answer here would make me more comfortable with the framing.
2.	How sensitive are the gains to the receptive field and design of the ShortConv anchor? Since the appendix notes that some synthetic tasks are arranged to match this locality structure, I would like a clearer sense of how robust the advantage is when that alignment is weaker.
3.	Can the authors give a more concrete breakdown of where PRISM clearly improves over the strongest linear baselines, beyond average rank summaries? A sharper accounting of wins, ties, and losses would help calibrate the practical advantage.
4.	What are the actual compute and memory costs of increasing refinement depth L during training and inference? This would help clarify the real tradeoff behind the higher-rank write path.
5.	If the authors do intend a more general sequence-modeling claim, even one additional non-recommendation benchmark or a more explicit discussion of why such evidence is currently missing would substantially strengthen the paper.

**Limitations:**

The limitations section is present and generally responsible. I still think it would be stronger if it were more explicit about the current evidence being concentrated in recommendation and about the possibility that the input-anchored proxy may fail in settings that require more genuinely state-dependent correction.

**Strengths And Weaknesses:**

Strengths
1.	The paper is built around a clear and fairly intuitive architectural idea. I found the write/forget decoupling story helpful: it gives the model a real design principle, rather than presenting the method as a loose collection of engineering choices.
2.	The efficiency motivation is reasonable and, within the comparisons the paper chooses, mostly convincing. In particular, the gap against explicit optimization-style baselines such as TTT appears meaningful, so the systems angle does not feel decorative.
3.	I also appreciated that the paper goes beyond a single summary table. The ablations on refinement depth, anchoring, gain prediction, and nonlinearities make it easier to see what is actually carrying the method.
4.	The synthetic and mechanistic sections are not definitive, but they do at least try to connect the observed behavior back to the modeling claim, which is better than leaving the architectural intuition completely untested.

Weaknesses
5.	My main reservation is about scope calibration rather than the core idea itself. The paper is framed in fairly broad sequence-modeling language, while the strongest empirical evidence is concentrated in recommendation. I do not think that makes the paper invalid, especially if recommendation is the intended application focus, but I do think the manuscript should be more careful about distinguishing what is shown from what is still speculative.
6.	Relatedly, some of the theoretical and motivational prose runs a bit ahead of the empirical support. Showing that the write path can realize higher-rank updates is useful, but that by itself is not the same as demonstrating that PRISM faithfully recovers the behavior of a more expensive iterative solver. I would have liked a slightly more restrained discussion of what the theory establishes and what remains an intuition.
7.	The synthetic probing setup is interesting, though somewhat favorable to the proposed mechanism. The appendix notes that some logical tasks are arranged so the relevant operands fall within the local receptive field of the ShortConv anchor. That does not invalidate the experiments, but it does mean the probing evidence should be interpreted more as supportive analysis than as a broad stress test.
8.	I was less bothered by the recommendation focus than by the way the claims are phrased. If the intended contribution is primarily a strong architecture for sequential recommendation, the empirical package is reasonably solid. If the intended claim is a broader advance in general generative sequence modeling, then the current validation is not yet wide enough. This is mostly a framing issue, but it matters.
9.	Presentation is decent overall, but not fully polished. The paper is readable and the main narrative comes through, yet several passages lean too heavily on slogan-like framing, and there are small quality-control issues that should have been caught in a final pass. For example, one figure misspells “parallel.” This is minor, but it contributes to the sense that the draft could use another careful edit.

---

> ### Author Rebuttal · Authors · 2026-03-30
>
> We thank the reviewer for the constructive feedback. Below, we address your concerns point by point.
>
> **1. Generalization Beyond Recommendation (W1, W4, Q1, Q5)**
> We initially focused on recommendation as user sequences' high entropy and diverse interests rigorously stress-test high-rank updates, challenging standard Rank-1 linear attention. To prove general applicability, we evaluated PRISM (~130M) on WikiText-103.
>
> PRISM significantly outperforms linear baselines (GLA, DeltaNet). Furthermore, our Hybrid (PRISM+TF) model surpasses the pure Transformer in Test PPL while maintaining competitive throughput after recent optimizations.
>
> **Table A: Language Modeling Performance (WikiText-103, ~130M Params)**
> | Rank | Model | Params | Test PPL ↓ | Test BPC ↓ | Relative to Transformer |
> | :--- | :--- | :--- | :--- | :--- | :--- |
> | 1 | Hybrid (PRISM+TF) | 137.40M | 34.10 | 5.058 | - |
> | 2 | Hybrid_DN (DN+TF) | 136.47M | 34.70 | 5.117 | -0.9% |
> | 3 | Transformer | 136.47M | 35.02 | 5.130 | baseline |
> | 4 | PRISM | 137.40M | 38.57 | 5.268 | - |
> | 5 | GLA | 136.47M | 42.37 | 5.405 | +21.0% |
> | 6 | DeltaNet | 136.47M | 42.93 | 5.424 | +22.6% |
>
> **Table B: Training Efficiency and Throughput**
> | Model | Steady Throughput (tok/s) | Epoch 1 Time | Steady Epoch | Total Train Time | Relative Speed |
> | :--- | :--- | :--- | :--- | :--- | :--- |
> | GLA | ~41,800 | 487.9s | ~352s | 1.02h | Fastest |
> | Transformer | ~41,500 | 377.6s | ~355s | 1.00h | -0.7% |
> | Hybrid_DN (DN+TF)| ~40,400 | 489.3s | ~365s | 1.05h | -3.3% |
> | DeltaNet | ~40,000 | 536.2s | ~367s | 1.07h | -4.3% |
> | PRISM | ~40,000 | 524.0s | ~367s | 1.07h | -4.3% |
> | Hybrid (PRISM+TF)| ~39,000 | 600.0s | ~376s | 1.10h | -6.0% |
>
> We will include these results in the revised manuscript to support our broader claims and adjust our theoretical framing accordingly.
>
> **2. Theoretical Claims, ShortConv Bias, and Synthetic Tasks (W2, W3, Q2)**
> To isolate the Rank-L write path's impact and eliminate local receptive field bias, we augmented all baselines (including LA and MoM) with the identical ShortConv module in synthetic tests.
>
> Since all models used identical ShortConv anchors, LA/MoM's failure and PRISM's success on logical tasks (e.g., XOR) definitively prove reasoning gains stem from high-rank iterative refinement, not the receptive field. Our ablation verifies the anchor provides stability, but the Rank-L solver unlocks expressivity. We will clarify this in Section 5.5.
>
> **3. Concrete Breakdown of Improvements (Q3)**
> To provide a sharper accounting of PRISM's advantages, we compared it against the strongest linear baselines across 4 datasets and 4 metrics (Hit@200, NDCG@200, Hit@500, NDCG@500), totaling 16 pairs per baseline.
>
> **Table C: Win/Tie/Loss Breakdown vs. Baselines**
> | Comparison | Win | Tie | Loss |
> | :--- | :--- | :--- | :--- |
> | PRISM vs. TITANS | 7 | 0 | 9 |
> | PRISM vs. Mamba2 | 15 | 1 | 0 |
> | PRISM vs. GDeltanet | 14 | 0 | 2 |
>
> We will add this explicit breakdown to the appendix. While PRISM may not strictly surpass explicit solvers like TITANS in all metrics, absolute metric dominance is not our goal. Instead, PRISM achieves high-fidelity approximation of these solvers while unlocking parallel training, delivering >100x speedups over TTT-class architectures.
>
> **4. Compute and Memory Costs of Refinement Depth (Q4)**
> We benchmarked training throughput for L=1 to 3 with a fixed token budget (batch_size × seq_len = 16K).
>
> **Table D: Training Throughput (kToken/s) vs. Refinement Depth (L)**
> | L | seq_len=2K | seq_len=4K | seq_len=8K | seq_len=16K | Relative to L=1 |
> | :--- | :--- | :--- | :--- | :--- | :--- |
> | L = 1 | 936.1 | 631.4 | 404.8 | 205.5 | 1.00× |
> | L = 2 (Ours) | 514.4 | 339.7 | 192.1 | 109.9 | ~0.54× |
> | L = 3 | 354.9 | 232.7 | 130.9 | 79.6 | ~0.38× |
>
> Throughput drops monotonically with L but scales linearly with sequence length, confirming overhead is confined to the SRAM-resident solver, preserving global O(N) complexity. We will add this to the appendix.
>
> **5. Presentation and Quality Control (W5)**
> We corrected the "parallel" typo in Figure 1 and other minor inconsistencies. We will also revise the prose to ensure the framing is more objective and precise.

---

> > ### Author Rebuttal · Reviewer_ar6Q · 2026-04-03
> >
> > Thank you for the rebuttal. I will keep my rating!

---

### Decision · Program_Chairs · 2026-04-30

**Decision:**

Accept (regular)

**Comment:**

The paper proposes a linear-time sequence modeling architecture designed to bridge the gap between the expressivity of explicit iterative optimization solvers and the hardware parallelism of linear sequence models.  The reinterpretation of linear attention as a one-step online optimization process cleanly identifies the rank-1 bottleneck, providing strong justification for the decoupled write/forget design. During the rebuttal, the authors effectively addressed the issues by providing competitive language modeling results  and benchmarking against optimized TTT baselines. Ultimately, while pure PRISM does not surpass optimized Transformers in all generative domains, its ability to outperform existing linear baselines while offering massive throughput advantages constitutes a significant contribution, provided the authors incorporate the rebuttal data, temper their broader claims, and clarify the limitations of their hardware profiling and memory proxy in the camera-ready version.